# Waning-and-waxing shape changes in ionic nanoplates upon cation exchange

Zhanzhao Li[1], Masaki Saruyama [1] ✉, Toru Asaka [2] & Toshiharu Teranishi [1] ✉

Flexible control of the composition and morphology of nanocrystals (NCs) over a wide range is an essential technology for the creation of functional nanomaterials. Cation exchange (CE) is a facile method by which to finely tune the compositions of ionic NCs, providing an opportunity to obtain complex nanostructures that are difficult to form using conventional chemical synthesis procedures. However, due to their robust anion frameworks, CE cannot typically be used to modify the original morphology of the host NCs. In this study, we report an anisotropic morphological transformation of $Cu_{1.8}S$ NCs during CE. Upon partial CE of $Cu_{1.8}S$ nanoplates (NPLs) with $Mn^{2+}$, the hexagonal NPLs are transformed into crescent-shaped $Cu_{1.8}S–MnS$ NPLs. Upon further CE, these crescent-shaped NPLs evolve back into completely hexagonal MnS NPLs. Comprehensive characterization of the intermediates reveals that this waxing-and-waning shape-evolution process is due to dissolution, redeposition, and intraparticle migration of $Cu^+$ and $S^{2-}$. Furthermore, in addition to $Mn^{2+}$, this CE-induced transformation process occurs with $Zn^{2+}$, $Cd^{2+}$ and $Fe^{3+}$. This finding presents a strategy by which to create heterostructured NCs with various morphologies and compositions under mild conditions.

Cation exchange (CE) is an active area of research as it allows us to precisely tune the composition of ionic nanocrystals (NCs) under mild conditions[1–5]. Accordingly, it has provided various classes of NCs with metastable crystal structures and/or multiple components that are difficult to produce through direct synthetic routes, thereby fostering advances in active catalysts and luminescent materials[6–9]. In general, CE reactions retain the overall morphology and crystal system of the host NC due to the rapid diffusion of small cations in the robust anion sublattices[10–12]. However, certain specific cases of CE, which induce large lattice-volume change, reduction of the formation energy of a crystal system, and/or increase of its surface energy, can cause alteration of the anion sublattice without transforming the overall morphology of the host NC[13,14].

Although it was widely believed that CE does not induce shape-transformation in host NCs, recent studies have revealed that CE can cause shape transformation in ionic NCs[15–17]. The formation of a hollow structure is a representative case. This can be explained by the nanoscale Kirkendall effect, in which different diffusion rates of the incoming and outgoing cations enforce anion migration to compensate for the total charges within an NC[18,19]. Specifically, CE of nanoplates (NPLs) can result in in-plane shape evolution, forming atypical structures with toroidal and biconcave shapes. Accordingly, studying the mechanisms behind CE-induced shape transformations can facilitate the synthesis of intricate functional nanomaterials that are not readily fabricated using conventional approaches[20,21].

In this study, we present a distinctive "waning and waxing" morphological evolution of ionic NCs during CE. By tuning the quantity of the foreign cation ($Mn^{2+}$), host hexagonal $Cu_{1.8}S$ NPLs undergo an evolution into crescent-shaped $Cu_{1.8}S–MnS$ heterostructured NPLs (HNPLs) during partial CE, subsequently reverting back to hexagonal MnS NPLs upon supplying adequate $Mn^{2+}$. Thorough characterization of the intermediate products in this waning-and-waxing process, we elucidated the underlying mechanism governing this drastic transformation. Partial CE with $Zn^{2+}$ also induces shape changes in NPLs, and a variety of metal cations can be utilized to repair chipped NPLs, resulting in the formation of various hexagonal HNPLs. Furthermore,

[1]Institute for Chemical Research, Kyoto University, Gokasho, Uji, Kyoto, Japan. [2]Division of Advanced Ceramics, Nagoya Institute of Technology, Nagoya, Aichi, Japan. ✉e-mail: saruyama@scl.kyoto-u.ac.jp; teranisi@scl.kyoto-u.ac.jp

such CE-induced transformation is also observed in rod-shaped $Cu_{1.8}S$ NCs. This discovery demonstrates the significant capability of CE reactions to not only substantially modify the compositions but also the morphologies of ionic NCs, expanding the morphological possibilities of CE products and providing an accessible methodology to create increasingly intricate nanostructures.

## Results

### $Mn^{2+}$ CE products of $Cu_{1.8}S$ NPLs

Roxbyite (r) $Cu_{1.8}S$ NPLs were synthesized as the host NCs for CE reactions in accordance with the reported procedure[22]. The transmission electron microscopy (TEM) images in Fig. 1a show hexagonal $Cu_{1.8}S$ NPLs with a uniform thickness and a diameter of $5.3 \pm 0.3$ nm and $70.0 \pm 3.4$ nm, respectively. The high-resolution TEM (HRTEM) images shown in Fig. 1b, c and Suppl. Fig. 1a, b reveal that the (400) and (008) planes are aligned along the vertical and horizontal directions of the plates, respectively.

CE of $Cu_{1.8}S$ NPLs with $Mn^{2+}$ was initiated by injecting a tri-$n$-octylphosphine (TOP) solution of $Cu_{1.8}S$ NPLs into a 1-octadecene (ODE) solution of an $MnCl_2$–oleylamine (OLAM) complex at 100 °C. Adding excess $Mn^{2+}$ precursor (molar ratio of $MnCl_2$ to $Cu_{1.8}S$, $[MnCl_2]/[Cu_{1.8}S] = 4$) enabled almost complete CE from $Cu_{1.8}S$ to MnS in 5 min, as shown by the energy-dispersive X-ray spectroscopy (EDX) results (Cu:Mn:S atomic ratio = 2:48:50; Fig. 1m). The CE reaction maintained the original plate-shape, producing NPLs with thickness × diameter dimensions of $5.2 \pm 0.4 \times 71.5 \pm 3.1$ nm (Fig. 1d). The X-ray diffraction (XRD) pattern shows the formation of a wurtzite (w)-MnS phase (Fig. 1l), and the HRTEM images reveal that the (002) and (110) planes of w-MnS are aligned along the vertical and horizontal directions of the NPLs, respectively (Fig. 1e, f and Suppl. Fig. 1c, d). These results indicate that the hexagonal-close-packed (hcp) $S^{2-}$ sublattice structure of the host r-$Cu_{1.8}S$ NPLs is retained during CE (Fig. 1g), as observed in our previous report[13]. This retention of shape and crystal structure highlights the robustness of the hcp $S^{2-}$ framework, and $Cu_{1.8}S$ and MnS NPLs are similar in size owing to the small lattice volume change (+1.8%) from r-$Cu_{1.8}S$ to w-MnS (Fig. 1g). The broad optical absorption peak at around 1900 nm, which is derived from the localized surface plasmon resonance (LSPR) in $Cu_{1.8}S$ NPLs, was not observed after CE, confirming the disappearance of the $Cu_{1.8}S$ phase[23] (Suppl. Fig. 2).

Interestingly, the quantity of $MnCl_2$ used has an impact on the morphology of the CE product. When using a sub-stoichiometric amount of $Mn^{2+}$ ($[MnCl_2]/[Cu_{1.8}S] <1$) for CE, incomplete crescent-shaped NPLs are formed (Fig. 1h–j). The EDX results shown in Fig. 1m reveal that significant amounts of Cu remain in these products, indicating that the CE terminates at intermediate stages. These partially cation-exchanged NPLs present XRD patterns assigned to both w-MnS and r-$Cu_{1.8}S$, suggesting the formation of $Cu_{1.8}S$–MnS heterostructures (Fig. 1l). The remaining LSPR absorption peaks also indicate the presence of $Cu_{1.8}S$ phase in the crescent-shaped NPLs (Suppl. Fig. 2).

During this shape transformation, the volumes of the individual NPLs also change. For instance, the original volume of the host $Cu_{1.8}S$ NPL is decreased by ≈27% at $[MnCl_2]/[Cu_{1.8}S] = 0.25$ (Fig. 1n). Considering the small lattice volume change from r-$Cu_{1.8}S$ to w-MnS (+1.8%), this drastic volume reduction is ascribed to the partial dissolution of the NPLs at the initial stage of CE. When the $[MnCl_2]/[Cu_{1.8}S]$ is adjusted to 1, most $Cu^+$ is replaced with $Mn^{2+}$ to form unchipped w-MnS NPLs, and the volumes of individual NPLs are very similar to those of the host NPLs (Fig. 1k, n).

These results imply that the progress of CE strongly affects the shape of the resulting NPL. To finely tune the CE conditions, we conducted successive CE reactions of $Cu_{1.8}S$ NPLs with a regulated $Mn^{2+}$ supply until the $[MnCl_2]/[Cu_{1.8}S]$ reaches ≈1 (0.10–0.13 for each injection; nine cycles in total). A sequence of TEM images of the products at each CE step highlights the step-by-step shape evolution in the waning-and-waxing process (Suppl. Fig. 3). With increasing the $Mn^{2+}$ supply, the volume of the individual NPLs initially decreases by ≈25% ($[MnCl_2]/[Cu_{1.8}S] = 0.42$), subsequently being restored to ≈99% of their original volume ($[MnCl_2]/[Cu_{1.8}S] = 1.0$) (Suppl. Fig. 3e). Because this experiment was conducted in a single-batch setup, the result suggests that the NPLs undergo a process involving early partial dissolution and subsequent regrowth to re-establish their hexagonal shape during the CE reaction.

### Characterization of crescent-shaped $Cu_{1.8}S$–MnS HNPLs

The structure of the crescent-shaped HNPLs formed at $[MnCl_2]/[Cu_{1.8}S] = 0.5$ was characterized by electron microscopy. The high-angle annular dark-field (HAADF)–scanning TEM (STEM) images in Fig. 2a show that the crescent-shaped $Cu_{1.8}S$–MnS HNPLs are composed of two distinguishable domains with a brighter 'crescent-string' region and a darker 'crescent-bow' region (see Fig. 2a for definitions of these terms). The STEM–EDX maps indicate that Cu is mainly located at the crescent-string region and Mn is distributed throughout the HNPLs (Fig. 2b–d). The STEM–EDX line profile from side view of the HNPLs reveals that Mn is present on both faces of the $Cu_{1.8}S$ NPL (Fig. 2e, f). The HRTEM image reveals that MnS layers of 1.5 nm thickness cover the $Cu_{1.8}S$ NPLs, indicating the formation of a $Cu_{1.8}S$@MnS core@shell structure (Fig. 2g). The STEM–EDX map from the top-view also indicates the presence of both MnS and $Cu_{1.8}S$ phases in the crescent-string region. The thickness of the $Cu_{1.8}S$ core is ≈5.0 nm, similar to that of the initial $Cu_{1.8}S$ NPLs (5.3 nm), indicating that the 1.5-nm MnS shells increase the total thickness of the HNPLs (≈8.0 nm). Figure 2h shows a structure model of the crescent-shaped $Cu_{1.8}S$–MnS HNPL based on these observations, wherein a chipped $Cu_{1.8}S$ NPL is embedded in an anisotropic MnS shell.

Crystallographic insight into the $Cu_{1.8}S$–MnS HNPLs was obtained through HRTEM observation. The HRTEM images and fast-Fourier-transform (FFT) pattern indicate heteroepitaxial connections between r-$Cu_{1.8}S$ and w-MnS (Fig. 2i–k). The interface model based on the parallel relationship between r-$Cu_{1.8}S$ [008] and w-MnS [110] indicates that the hcp $S^{2-}$ sublattice continuously connects at the heterointerface of the two phases (Fig. 2l). These results indicate that the overall shape transformation occurs while conserving the single-crystallinity of the host $Cu_{1.8}S$ NPLs.

### Waning mechanism of $Cu_{1.8}S$ NPLs

To better understand how the $Cu_{1.8}S$ NPLs are transformed during partial CE, the morphological evolution of the NPLs was carefully monitored by characterizing intermediate products during partial CE (i.e., $[MnCl_2]/[Cu_{1.8}S] = 0.5$). The EDX and XRD results confirm the temporal change of both the composition and the crystal structure of the products during CE (Suppl. Fig. 4). The TEM images in Fig. 3a–j reveal a shape change from hexagonal $Cu_{1.8}S$ NPLs to crescent-shaped $Cu_{1.8}S$–MnS HNPLs in just 5 min. In detail, the NPLs have a large basal area but have begun to chip from one side at 10 s (Fig. 3a), generating NPLs with non-uniform thickness, i.e., one side of ≈3.0 nm and the other of ≈9.0 nm (Fig. 3f and Suppl. Fig. 5). The top-view STEM–EDX map of an NPL at 10 s shows that Mn is concentrated in half the area of the NPL, while Cu is distributed throughout the whole area (Fig. 3k–m). The HRSTEM images in Fig. 3n indicate that the thin part is composed of r-$Cu_{1.8}S$-phase material. STEM–EDX maps from the side-views in Fig. 3o–q show that Cu is distributed throughout both the thick and thin parts, and Mn is mainly located as thin layers in the thick Cu part. These observations indicate that the intermediate heterostructures formed at 10 s are composed of thick $Cu_{1.8}S$@MnS core@shell structures and thin $Cu_{1.8}S$ species. At 30 s after CE starts, the $Cu_{1.8}S$ region not covered by MnS layers is observed to have thinned to ≈2.0 nm or partially broken, while the $Cu_{1.8}S$@MnS region has thickened to ≈9.5 nm (Fig. 3g and Suppl. Fig. 5), suggesting further intraparticle ion migration. The thin $Cu_{1.8}S$ parts are no longer evident at 1 min, and the resulting half-NPLs are observed to have deformed into a crescent-shape at 5 min (Fig. 3c–e).

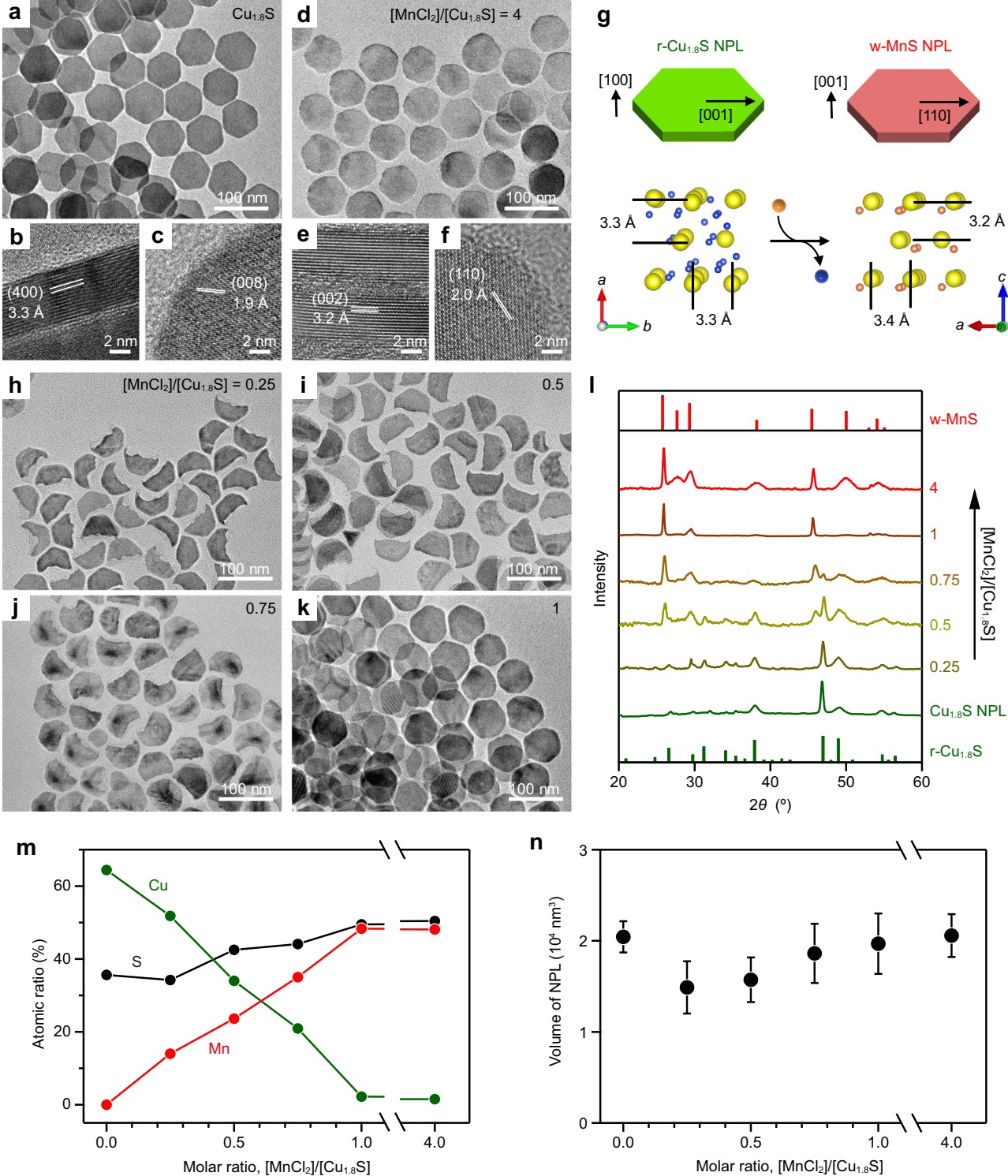

**Fig. 1 | CE of Cu₁.₈S NPLs with Mn²⁺ using different [MnCl₂]/[Cu₁.₈S].** a, d TEM and (b, c, e, f) HRTEM images of (a–c) host $Cu_{1.8}S$ NPLs and (d–f) MnS NPLs formed by CE of $Cu_{1.8}S$ NPLs with $Mn^{2+}$. b, e Side views and (c, f) top views. g Crystal structures of r-$Cu_{1.8}S$ NPLs and w-MnS NPLs. h–k TEM images of CE products at [MnCl₂]/ [Cu₁.₈S] = (h) 0.25, (i) 0.5, (j) 0.75, and (k) 1. l XRD patterns, (m) EDX results (green: Cu, red: Mn, black: S), and (n) volumes of individual NPLs obtained upon partial CE at various [MnCl₂]/[Cu₁.₈S]. Reference XRD patterns: w-MnS (red; ICCD 01-089-4089) and r-$Cu_{1.8}S$ (green; ICCD 00-064-0278). Error bars represent standard deviations calculated from standard deviations of base areas ($n = 20$) and thicknesses ($n = 20$).

It should be noted that a partial dissolution of the NPLs takes place, causing a volume reduction during the waning process (Fig. 1n). X-ray fluorescence (XRF) spectroscopy revealed a considerable amount of S in conjunction with Cu in the supernatant during the purification of the crescent-shaped Cu₁.₈S–MnS HNPLs, indicating that

partial dissolution of $Cu_{1.8}S$ occurs in the partial CE process (Suppl. Fig. 6).

When a control reaction was conducted without MnCl₂, the original hexagonal morphology, crystal structure, and volume of the Cu₁.₈S NPLs were largely retained (Suppl. Fig. 7a, b, e),

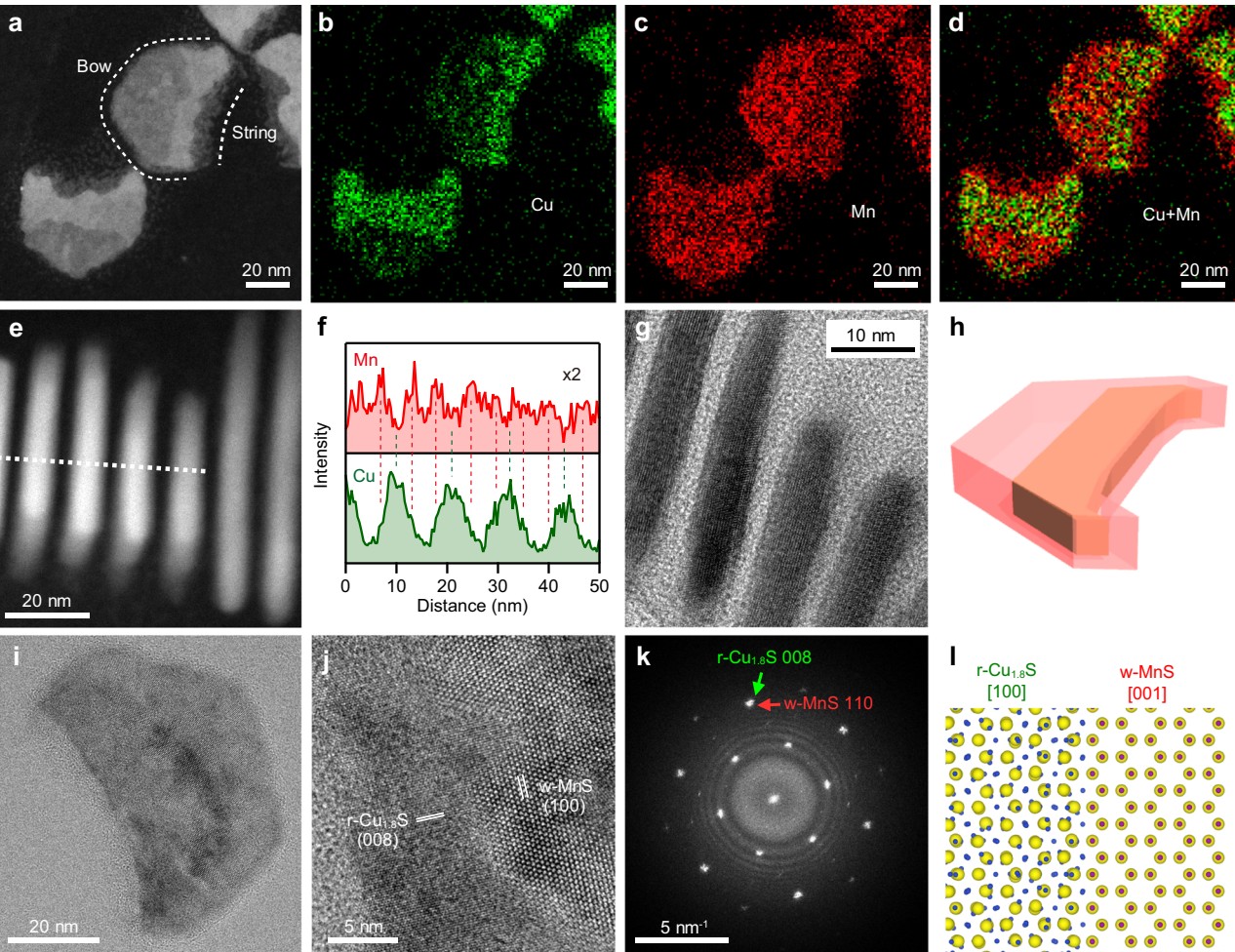

**Fig. 2 | Structure characteristics of crescent-shaped Cu₁.₈S–MnS HNPLs. a** Top-view STEM–HAADF and (**b–d**) STEM-EDX maps (green: Cu-K, red: Mn-K) of Cu₁.₈S–MnS HNPLs. **e** Side-view HAADF–STEM image of Cu₁.₈S–MnS HNPLs and (**f**) STEM–EDX line profile along the dotted line in **e**. Green: Cu, red: Mn. **g** Side-view HRTEM image of Cu₁.₈S–MnS HNPLs. **h** Structure schematic of Cu₁.₈S–MnS HNPLs. **i, j** Top-view HRTEM images of Cu₁.₈S–MnS HNPLs. **k** FFT pattern of **i**. **l** Crystal model of the epitaxial Cu₁.₈S/MnS heterointerface.

suggesting that OLAM and TOP are not responsible for the dissolution of the NPLs. When $MnCl_2$ was substituted with tetraoctylammonium (TOA) chloride, which does not induce CE but supplies $Cl^-$, round-edged $Cu_{1.8}S$ NPLs devoid of the vertexes in the original hexagonal $Cu_{1.8}S$ NPLs were obtained (Suppl. Fig. 7c)[24]. Although the crystal structure of $r\text{-}Cu_{1.8}S$ was maintained, the volume of the individual NPLs decreased by ≈17%, indicating etching by $Cl^-$ (Suppl. Fig. 7d, e).

Based on the above characterization, we speculate the transformation mechanism of the waning process. The formation of NPLs with non-uniform thickness in the early stage (e.g., ≈3 nm and ≈9 nm at 10 s) from the flat $Cu_{1.8}S$ NPLs (5.3 nm) is likely initiated by anisotropic intraparticle ion migration within individual NPLs after the CE with $Mn^{2+}$ from one side of $Cu_{1.8}S$ NPLs[15,16,25]. Initiation of CE from a single location on a NC forms the starting point for the subsequent anisotropic CE, often leading to the formation of Janus-type heterostructure, as observed in many cases[8,26–34]. Explanations for this phenomenon have been often provided by the formation of a crystallographically stable heterointerface[31,33] and/or the presence of a high activation energy for the CE reaction[30,34]. These explanations would also apply to our case, where CE of the $Cu_{1.8}S$ NPL with $Mn^{2+}$ started from a single location. Subsequently, the imbalance between the

rapid outward diffusion of the host $Cu^+$ and the slow inward diffusion of the guest $Mn^{2+}$ (as shown later) causes the anisotropic shape transformations[16]. Such a transformation triggered by a large difference in inward/outward cation migration rates has been shown in several cases, which are often explained as nanoscale Kirkendall effect[15–18]. In the case of NPLs, ring[15,18] and biconcave-shaped[16] nanostructure have been obtained through intraparticle ion migration in in-plane direction during CE initiated from all edges of the NPLs. In our case, the progress of CE with $Mn^{2+}$ from one side of $Cu_{1.8}S$ NPLs causes the directional in-plane ion migration, leading to the formation of anisotropic NPLs with non-uniform thickness. The CE continuously propagates MnS phases in NPLs from the edge, which is evidenced by the position of $Cu_{1.8}S$ phases within the crescent-shaped NPLs, not near the edge but towards the center (Fig. 2).

In addition to the anisotropic Kirkendall-type intraparticle ion migration, the decomposition of $Cu_{1.8}S$ NPLs triggered by strong coordination between $Cu^+$ and $Cl^-$ promotes the large deformation. Once partial CE with $Mn^{2+}$ occurs at a single location of NPL, interparticle diffusion of ions creates structural defects on the opposite side of NPL to provide exposed fresh and unstable $Cu_{1.8}S$ surface as the starting point for accelerated etching by $Cl^-$ (Suppl. Fig. 8). On such a highly reactive surface, TOP is also expected to act as a supplemental

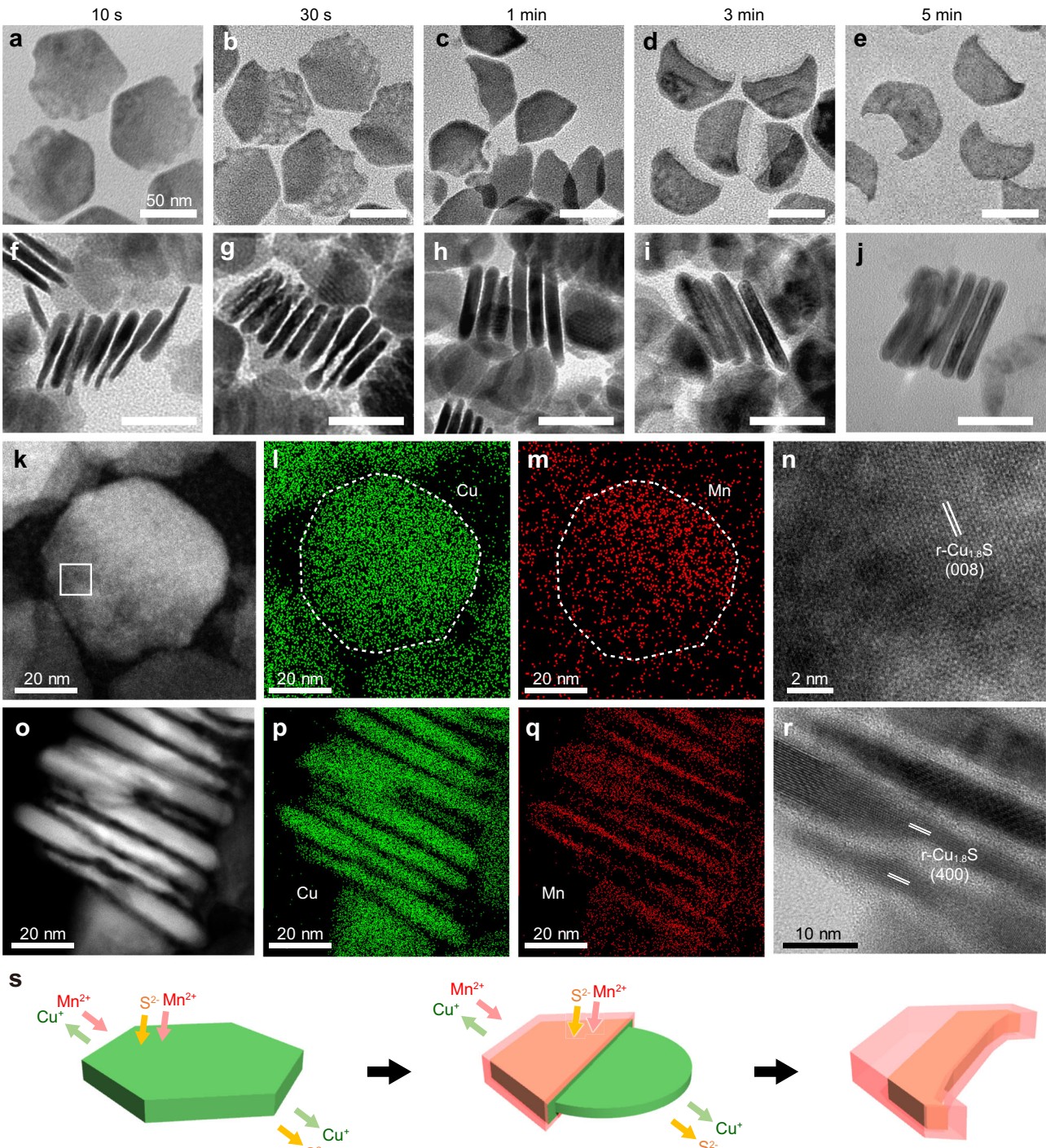

**Fig. 3 | 'Waning' of Cu₁.₈S NPLs into crescent-shaped Cu₁.₈S–MnS HNPLs.**
**a–j** TEM images (**a–e**: top views; **f–j**: side views) of Cu₁.₈S–MnS HNPLs formed during partial CE with $Mn^{2+}$ (reaction condition: [MnCl₂]/[Cu₁.₈S] = 0.5) at (**a, f**) 10 s, (**b, g**) 30 s, (**c, h**) 1 min, (**d, i**) 3 min and (**e, j**) 5 min. Scale bars = 50 nm. **k** Top-view HAADF–STEM image and (**l, m**) STEM–EDX maps for (**l**) Cu-K (green) and (**m**) Mn-K (red) in Cu₁.₈S–MnS HNPLs at 10 s. Dotted lines outline an HNPL. **n** Magnified STEM image of the selected area in **k**. **o** Side-view HAADF–STEM image and (**p, q**) STEM–EDX maps for (**p**) Cu-K (green), (**q**) Mn-K (red), and (**r**) HRTEM image of Cu₁.₈S–MnS HNPLs at 10 s. **s** Schematic of the waning process in (H)NPLs.

etching agent for $S^{2-}$, further accelerating NPLs dissolution[35]. The exposed Cu₁.₈S not covered by a MnS shell in the intermediate (observed at 10 s) is susceptible to etching and subsequently disappears (after 1 min, Fig. 3c).

Another plausible reaction in the waning process is the MnS deposition on NPLs. After the Cu₁.₈S NPLs is partially etched, the dissolved $S^{2-}$ reacts with $Mn^{2+}$ to cause the growth of MnS on residual Cu₁.₈S (as shown later)[36]. Considering that the CE generally proceeds

from an edge of Cu₁.₈S NPLs, the MnS shells on both faces of Cu₁.₈S NPLs might grow through this MnS deposition mechanism (Suppl. Fig. 8). These results indicate that, after the CE initiates, the kinetic balance between anisotropic CE progression, Cu₁.₈S etching and MnS deposition induces the specific transformation of hexagonal NPLs into crescent-shaped HNPLs, as summarized in Fig. 3s.

To investigate the generality of the waning phenomenon observed in Cu₁.₈S NPLs, we replaced MnCl₂ with other metal chlorides

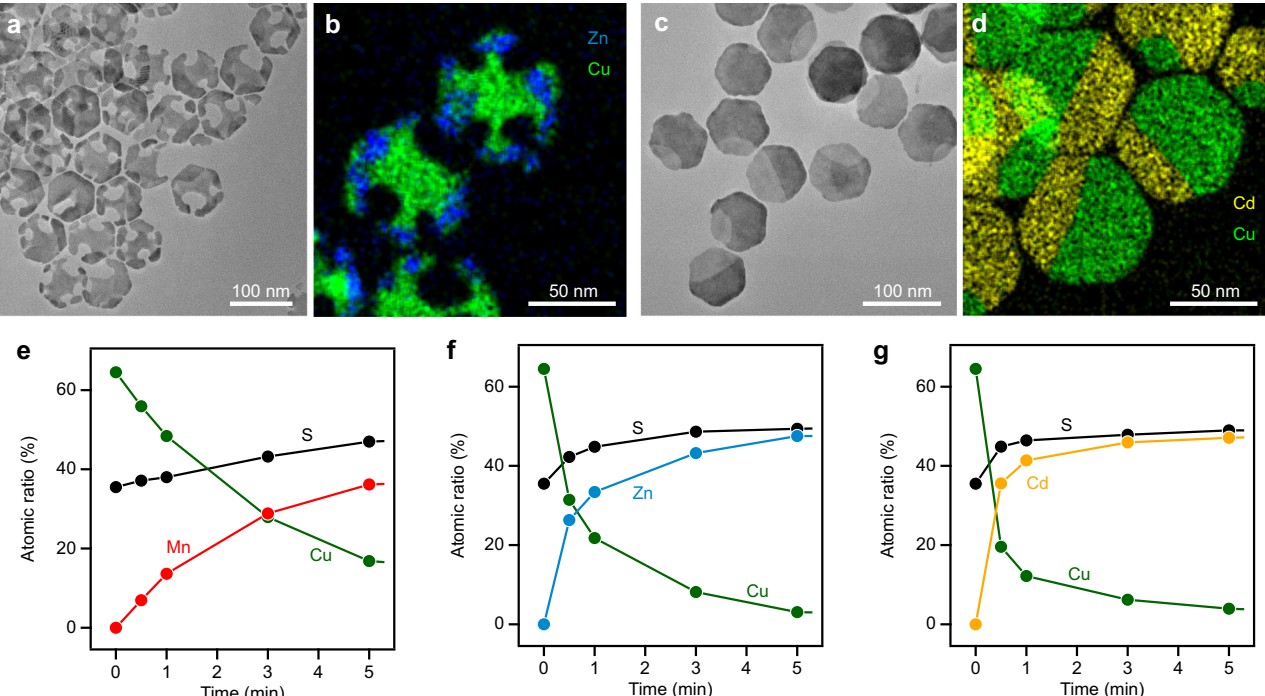

**Fig. 4 | Partial CE with different metal chlorides. a** TEM image and (**b**) STEM−EDX map (green: Cu-K, blue: Zn-K) of chipped $Cu_{1.8}S$−ZnS HNPLs. **c** TEM image and (**d**) STEM−EDX map (green: Cu-K, yellow: Cd-K) of Janus-like $Cu_{1.8}S$−CdS HNPLs.

Reaction conditions: $[MCl_2]/[Cu_{1.8}S] = 0.5$ (M = Zn or Cd). Temporal EDX results for CE of $Cu_{1.8}S$ NPLs with (**e**) $Mn^{2+}$, (**f**) $Zn^{2+}$ and (**g**) $Cd^{2+}$. Reaction conditions: $[MCl_2]/[Cu_{1.8}S] = 1$ (M = Zn or Cd) at 80 °C.

and ascertained the impact of metal species. As observed by TEM and STEM−EDX (Fig. 4a, b), partial CE with $Zn^{2+}$ results in the formation of chipped $Cu_{1.8}S$−ZnS HNPLs with open holes, where Zn is concentrated on the periphery and Cu in the central region. The volume of the host NPLs decreased by ≈27%, and the formation of holes was attributed to severe dissolution of the $Cu_{1.8}S$ phases, as evidenced by HRTEM images (Suppl. Fig. 9). These findings indicate that CE with $Zn^{2+}$ initiates at the corners of $Cu_{1.8}S$ NPLs while the $Cu_{1.8}S$ phases undergoes partial dissolution. Conversely, partial CE with $Cd^{2+}$ results in the formation of hexagonal Janus-like $Cu_{1.8}S$−CdS HNPLs without shape transformation (Fig. 4c, d).

The difference in the shapes of products obtained by partial CE with distinct metal ions can be attributed to differences in CE rate. CE reactions that induce morphological changes are typically triggered by a significant disparity in the diffusion rates between host and guest cations[17,18]. For example, Lee et al. controlled the degree of Kirkendall effect by tuning the mutual diffusion rates of $Cu^+$ and $In^{3+}$ in CE from $Cu_{3-x}P$ to InP[17]. They showed that rapid outward diffusion of $Cu^+$ and sluggish inward diffusion of $In^{3+}$ drive the migration and dissolution of anions, resulting in the formation of cracks and voids in NPLs. Similar Kirkendall-type shape evolution has been observed for hollow $CuInS_2$ NCs[18].

The CE rates for different guest cations ($M^{2+}$; M = Mn, Zn or Cd) were compared by monitoring the temporal evolution of Cu:M:S mole ratio in cation exchanged products using EDX analyses ($[MnCl_2]/[Cu_{1.8}S] = 1$). CE reactions were carried out at a lower temperature (80 °C) to suppress the excessively rapid CE reactions that occur at a more typical temperature (100 °C) that hinder accurate comparison. The EDX results shown in Fig. 4e−g indicate that the CE rates decrease in the order $Cd^{2+} > Zn^{2+} > Mn^{2+}$, which can be qualitatively rationalized by the estimated Gibbs free energy changes for CE reactions of $Cu_{1.8}S$ with these metal cations (Suppl. Table 1)[3]. While the rapid inward diffusion of guest cations ($Cd^{2+}$) preserves the parent nanostructure by balancing the rapid extraction of $Cu^+$ by CE, the substantial difference in diffusion rates between the host ($Cu^+$, faster) and guest ($Zn^{2+}$ and

$Mn^{2+}$, slower) cations drives migration and leads to Kirkendall-type morphological evolution of NC[19]. Such anion migration causes damage to the $Cu_{1.8}S$ phase, forming less-stable surfaces that are susceptible to etching, promoting the drastic shape changes in CE reactions with $Mn^{2+}$ and $Zn^{2+}$.

## Waxing mechanism of $Cu_{1.8}S$−MnS HNPLs

As observed in the successive CE reactions, the crescent-shaped $Cu_{1.8}S$−MnS HNPLs revert back to the original hexagonal NPLs upon complete CE (Suppl. Figure 3). To investigate this waxing process, we introduced additional $Mn^{2+}$ ($[MnCl_2]/[Cu_{1.8}S−MnS] = 4$) into the reaction solution containing the crescent-shaped $Cu_{1.8}S$−MnS HNPLs ($[MnCl_2]/[Cu_{1.8}S−MnS] = 0.5$) and identified the products at various reaction times (Fig. 5). CE restarts immediately after additional $Mn^{2+}$ injection (Suppl. Fig. 10), and the crescent-shaped HNPLs are transformed into hexagonal NPLs within 5 min (Fig. 5a–h). Within 30 s, the missing parts of the $Cu_{1.8}S$−MnS HNPLs are partially repaired (Fig. 5a). STEM−EDX mapping revealed that the partially repaired regions contain Mn but not Cu, while both elements are detected in the original crescent-shaped parts (Fig. 5i, j). The HRTEM image indicates that the repaired region has a non-crystalline character, unlike the original crescent part (Fig. 5k–m), suggesting that the NPLs are restored through the growth of amorphous MnS[37,38]. Additionally, the thickness of the HNPLs decreases from 8.5 to 6.9 nm in the first 1 min (Suppl. Fig. 11). The side-view STEM−EDX map of the HNPLs formed at 30 s shows that Cu is exclusively located on one HNPL face (Fig. 5n, o). The HRTEM image also confirms the compositionally anisotropic nature of the r-$Cu_{1.8}S$/w-MnS heterostructures, where the r-$Cu_{1.8}S$ (400) and w-MnS (002) planes aligned parallel to the thickness direction (Fig. 5p). This asymmetric heterointerface formation may be attributed to the distinct polarities between the 001 and 00$\bar{1}$ surfaces of w-MnS[33]. The migration of $Cu^+$ in the $Cu_{1.8}S$@MnS core@-shell regions likely occurs towards one surface of the NPLs to form the most stable r-$Cu_{1.8}S$/w-MnS heterointerface. Furthermore, the

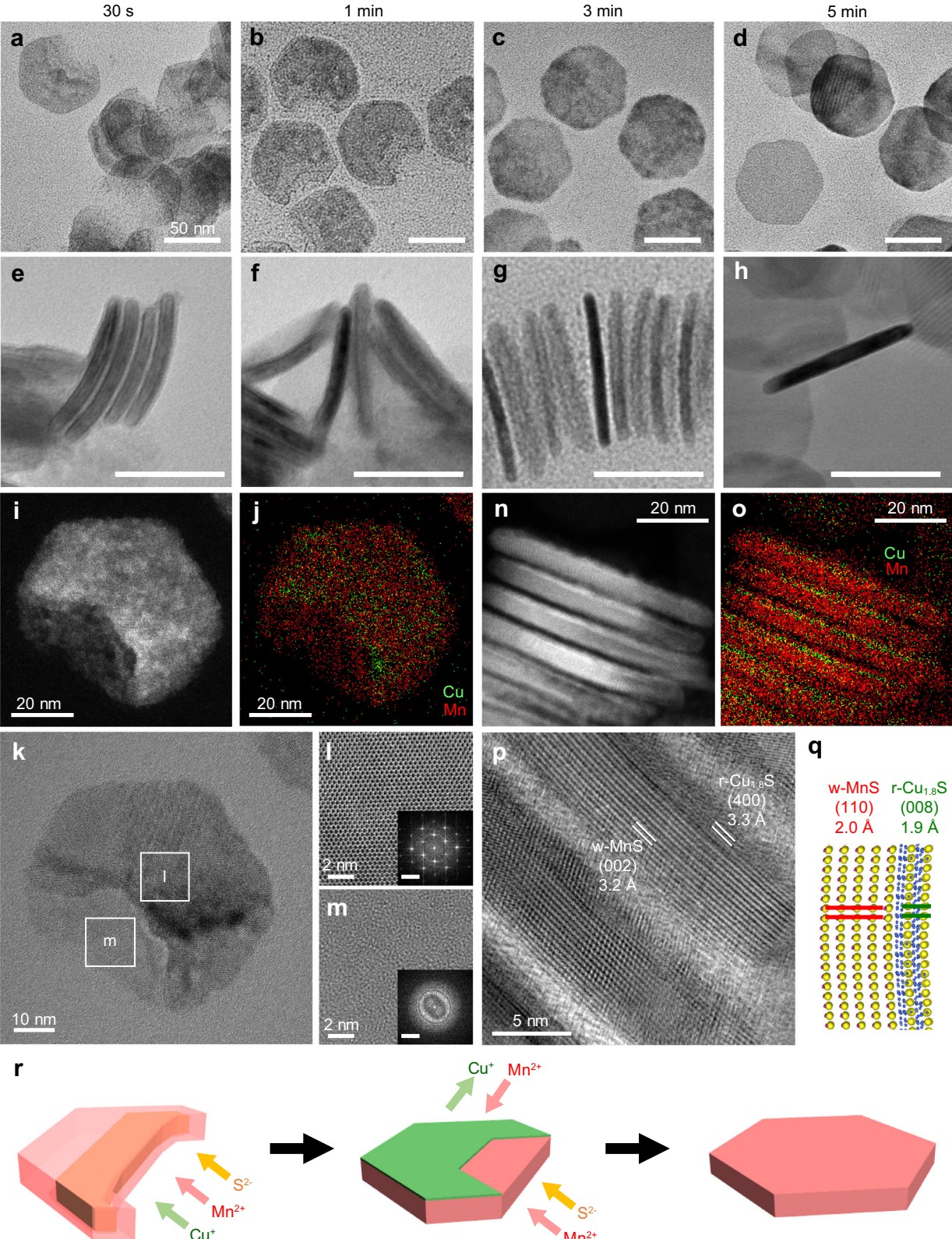

**Fig. 5 | 'Waxing' of crescent-shaped Cu₁.₈S−MnS HNPLs into complete hexagonal MnS NPLs. a−h** TEM images (**a−d**: top views; **e−h**: side views) of (H)NPLs formed at (**a, e**) 30 s, (**b, f**) 1 min, (**c, g**) 3 min and (**d, h**) 5 min after injecting additional Mn²⁺ (Reaction conditions: [MnCl₂]/[Cu₁.₈S−MnS] = 4). Scale bars = 50 nm. **i, n** HAADF−STEM and (**j, o**) STEM−EDX maps (green: Cu-K, red: Mn-K) and (**k, p**) HRTEM images of HNPLs formed at 30 s after MnCl₂ injection. **i−k** Top-views and (**n−p**) side-views. **l, m** Magnified HRTEM images of the selected area in **k**. Insets in (**l, m**) show corresponding FFT patterns (scale bars = 5 nm⁻¹). **q** Schematic structure of bent Cu₁.₈S−MnS HNPLs. **r** Schematic showing the waxing of Cu₁.₈S−MnS HNPLs.

anisotropic strain resulting from the lattice mismatch between r-$Cu_{1.8}S$ (008) (0.19 nm) and w-MnS (110) (0.20 nm) at the hetero-interface causes bending of the NPLs to form curved structures (Fig. 5e, f, q)[39].

Because the transformation from the crescent-shaped to the intermediate NCs with bilayer structure (at 30 s) involves a large structural change, there should be other intermediate structures in the earlier stage. However, the waxing stage progresses quite rapidly, making it difficult to experimentally capture fine intermediate snapshots. In previous works, reconstruction of heterointerfaces in NCs has been often observed in partial CE reactions. The large mobility of cations in $Cu_{1.8}S$ especially under heated conditions can rearrange two distinct domains with multiple patchy structure[40] or core-shell structure[31] into a smaller number (area) of heterointerface. These works suggest that, in our case, the $Cu_{1.8}S$–MnS bilayer structure is spontaneously formed by generating more thermodynamically stable heterointerface from the $Cu_{1.8}S$@MnS core@shell structure during CE progression[41].

At 3 min, the composition, shape, and crystal structure of the HNPLs is nearly identical to those of MnS NPLs obtained through the one-step CE at $[MnCl_2]/[Cu_{1.8}S]$ = 4 (Fig. 5c, g and Suppl. Fig. 12). The HRTEM images and FFT pattern show the single-crystalline nature of MnS NPLs, indicating amorphous MnS recrystallizes in this period (Suppl. Fig. 12). At 3 min, the thickness of the NPLs is decreased to 5.3 nm, which is almost equivalent to that of the host $Cu_{1.8}S$ NPLs (Suppl. Fig. 11). Prolonged CE for 5 min yields MnS NPLs with highly flat surfaces due to the release of $Cu_{1.8}S$/MnS interfacial strain by the disappearance of r-$Cu_{1.8}S$ layers (Fig. 5d, h). Figure 5r illustrates transforming the crescent-shaped $Cu_{1.8}S$–MnS HNPLs into complete hexagonal MnS NPLs in waxing process based on the microscopic observation.

The volume recovery observed in the waxing process cannot be solely explained by progression of the CE, because the lattice volumes of r-$Cu_{1.8}S$ and w-MnS differ by only 1.8%. To fully restore the lost volume in the waning process, dissolved $S^{2-}$ should be utilized to form MnS phases. Indeed, the XRF results show that S is almost absent from the supernatant in the purification of the waxed MnS NPLs (Suppl. Fig. 13), suggesting that the waxing process consumes dissolved $S^{2-}$ from the reaction solution. To investigate this further, we conducted a control experiment in which $MnCl_2$ was introduced into the OLAM/TOP/ODE solution containing "purified" crescent-shaped $Cu_{1.8}S$–MnS HNPLs to exclude dissolved $S^{2-}$ from the reaction solution. Consequently, incomplete-hexagonal MnS NPLs with volumes ≈25% less than those of the complete hexagonal MnS NPLs were obtained (Suppl. Fig. 14). This result strongly suggests that dissolved $S^{2-}$ anions in the partial CE process act as a $S^{2-}$ source for the waxing of the crescent-shaped HNPLs into hexagonal NPLs, in addition to the CE of the remaining $Cu_{1.8}S$ portion with $Mn^{2+}$.

A distinctive phenomenon in the waxing process is the decrease in the NPL thickness when the $Cu_{1.8}S$-MnS bilayer structure (7.5 nm at 10 s) is transformed into MnS NPL (5.2 nm). We suggest two possible scenarios of how the thickness of the NPL reverts back to the original thickness. The first is based on the reconstruction process after the CE reaction. If the CE proceeds in the $Cu_{1.8}S$ layer of $Cu_{1.8}S$–MnS bilayer structure, a thick MnS plate will form exclusively in that region, resulting in the formation of MnS NPLs with uneven thickness (Suppl. Fig. 15a). Considering the formation of flat, thin MnS NPLs, the shape reconstruction should take place to make the thickness uniform after the CE. Completely flat NPLs seem more stable due to the reduced surface energy than those with uneven thickness. In another scenario based on the etching and deposition process, the $Cu_{1.8}S$ layer of $Cu_{1.8}S$–MnS bilayer structure is rapidly etched by $Cl^-$ and/or TOP, leaving a thin MnS layer and releasing $S^{2-}$

as the precursor for MnS growth (Suppl. Fig. 15b). Because the thickness of MnS parts in bilayer structure (at 30 s in waxing process) is slightly thinner (≈4.7 nm) than that of final MnS NPLs (5.2 nm) (Suppl. Fig. 16), the two scenarios may occur simultaneously rather than just one process or the other.

## Waxing transformation with various metal cations

Considering that the recovery process proceeds through CE of the $Cu_{1.8}S$ remaining in the HNPLs and recrystallization of the metal sulfide from dissolved $S^{2-}$ and introduced metal cations, it may be hypothesized that different metal cations could also be employed to refill the vacancies in the hexagonal NPLs. Accordingly, we introduced different metal cations ($[M]/[Cu_{1.8}S]$ = 1; M = $Cd^{2+}$, $Zn^{2+}$ or $Fe^{3+}$) into the reaction solution after the formation of $Cu_{1.8}S$–MnS HNPLs through partial CE ($[MnCl_2]/[Cu_{1.8}S]$ = 0.5), resulting in successful restoration of the chipped HNPLs into complete hexagonal-shaped HNPLs (Fig. 6a–c). In the case of $Cd^{2+}$, the missing parts are filled with a w-CdS phase to produce Janus-like MnS–CdS HNPLs (Fig. 6g, m and Suppl. Figs. 17a–c and 18a). In the case of $Zn^{2+}$, the Mn and Zn segments are not clearly segregated in the HNPLs, as evidenced by the STEM–EDX map (Fig. 6h, n and Suppl. Fig. 17d–f). It is possible that $Mn^{2+}$ and $Zn^{2+}$ are partially mixed during the waxing process to give a w-$Mn_{1-x}Zn_xS$ solid-solution phase, as indicated by the XRD pattern (Suppl. Fig. 18b). CE with $Fe^{3+}$ results in the formation of hexagonal HNPLs containing three distinct regions (Fig. 6i, o and Suppl. Fig. 17g–j). In addition to MnS, the formation of a chalcopyrite $CuFeS_2$ phase was confirmed by XRD (Suppl. Fig. 18c). The STEM–EDX map reveals the presence of another region mainly containing Mn and Fe, suggesting the formation of a w-$Mn_{1-x}Fe_xS$ phase, as supported by the shifted XRD peaks of w-MnS (Suppl. Fig. 18c). The distinct spatial distributions of Mn and M within the HNPLs may arise from differences in the diffusion behavior of respective metal cations and the formation energy of mixed metal sulfide phases[42].

Furthermore, the waxing process was also found to be applicable to chipped $Cu_{1.8}S$–ZnS HNPLs. Further CE of the chipped $Cu_{1.8}S$–ZnS HNPLs with $Zn^{2+}$ results in filling of the voids, leading to the formation of complete hexagonal w-ZnS NPLs (Suppl. Fig. 19). Similar to the $Cu_{1.8}S$–MnS HNPL case, further addition of different metal cations, such as $Mn^{2+}$, $Cd^{2+}$ and $Fe^{3+}$ also transform the chipped $Cu_{1.8}S$–ZnS HNPLs into nearly hexagonal ZnS–CdS, ZnS–MnS and ZnS–$CuFeS_2$ HNPLs, respectively (Fig. 6d–f). The ZnS segments are located near the peripheries of the HNPLs, and the XRD patterns of w-ZnS and other sulfide phases are well-separated (Fig. 6j–l, p–r and Suppl. Figs. 20 and 21), indicating that the initially-formed ZnS portions are preserved during additional CE.

The synthesis of these multi-component NPLs has been accomplished through sequential CE with different cations[32,42]. Their formation mechanism has previously been regarded as a straightforward process involving initial partial CE followed by subsequent CE of the unreacted region. However, our results imply that, in specific cases, such CE processes involve a regenerable transformation through partial dissolution and recrystallization.

## $Mn^{2+}$ CE of $Cu_{1.8}S$ nanorods

The fact that the CE reactions occur in any shapes of NCs implies that the waning-and-waxing process can also take place in NCs with diverse morphologies. To demonstrate extensive adaptability of this CE-induced shape transformation, we adopted $Cu_{1.8}S$ nanorods (NRs; 57 nm length × 20 nm diameter). To examine the morphological changes at different stages of CE, we conducted a successive CE of $Cu_{1.8}S$ NRs with $Mn^{2+}$ until the $[MnCl_2]/[Cu_{1.8}S]$ reached ≈1. A series of TEM images was taken to determine the effect of the $MnCl_2/Cu_{1.8}S$ mole ratio on the shapes of the NRs (Suppl. Fig. 22a). The length of the NRs decreases to 45 nm until $[MnCl_2]/[Cu_{1.8}S]$ reaches ≈0.5, and it then

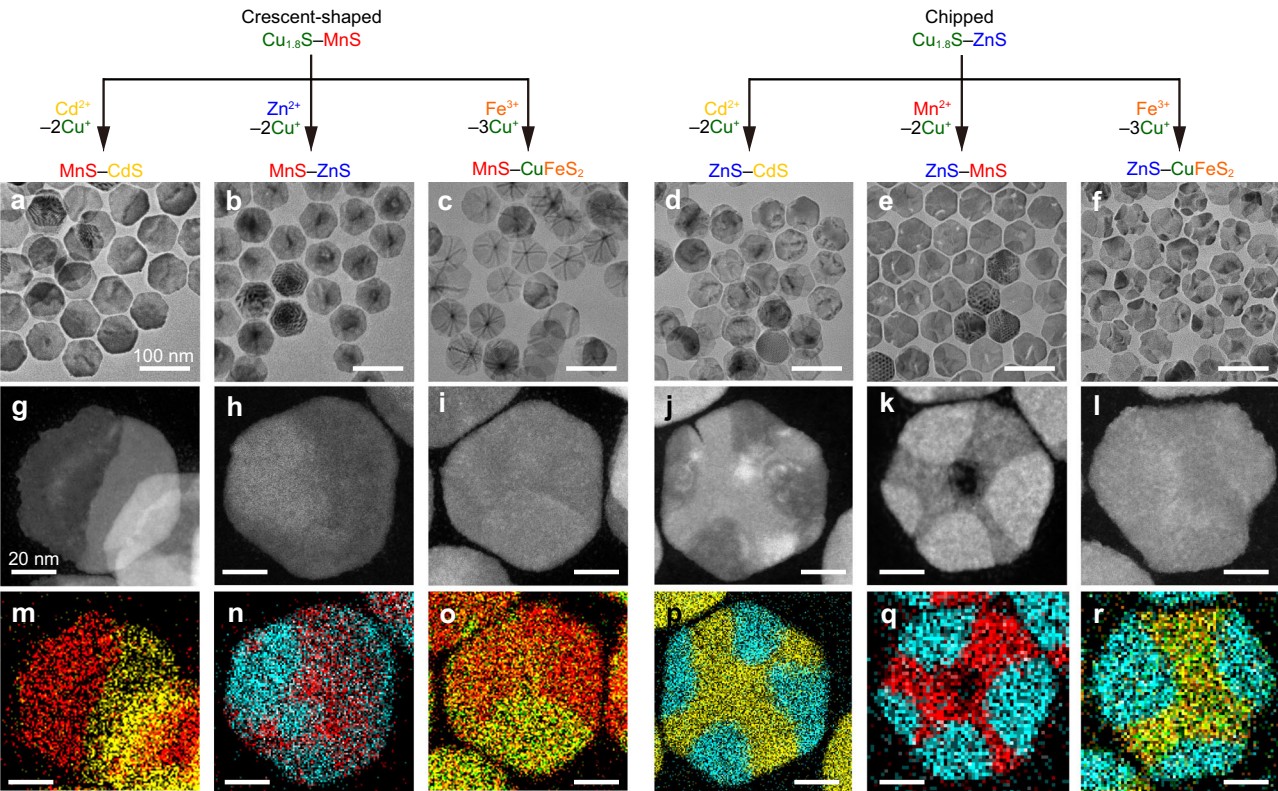

**Fig. 6 | Various HNPLs generated by waxing shape evolution. a–f** TEM images (scale bars = 100 nm), (**g–l**) HAADF–STEM images (scale bars = 20 nm) and (**m–r**) STEM–EDX maps (scale bars = 20 nm) of (**a, g, m**) MnS–CdS HNPLs, (**b, h, n**) MnS–ZnS HNPLs, (**c, i, o**) MnS–CuFeS$_2$ HNPLs, (**d, j, p**) ZnS–CdS HNPLs, (**e, k, q**) ZnS–MnS HNPLs, and (**f, l, r**) ZnS–CuFeS$_2$ HNPLs (red: Mn-K, yellow: Cd-K, blue: Zn-K, orange: Fe-K).

increases back to 54 nm upon further CE, while the NR diameter remains almost constant throughout the CE process (Suppl. Fig. 22b–d). This demonstrates that the waning-and-waxing process takes place upon CE of these NRs. It is noteworthy that the shape evolution proceeds along the [080]/[008] (in-plane of the NPL) or [400] (long-axis direction of the NR) of the r-Cu$_{1.8}$S NPLs or NRs, respectively. Since these planes (the edges of the NPLs and the tips of the NRs) are highly reactive owing to decreased ligand passivation, CE preferentially starts at these specific planes[13]. In fact, TEM images show the segregation of Cu$_{1.8}$S and MnS phases along the long axis of the NRs (Suppl. Fig. 22a), demonstrating that the waning-and-waxing process occurs along the direction of the CE reactions. This waning-and-waxing process offers additional prospects for shape modification of ionic NCs through CE protocols.

## Discussion
In summary, we have demonstrated anomalous waning-and-waxing shape evolution of Cu$_{1.8}$S NPLs during CE reactions. The initially hexagonal Cu$_{1.8}$S NPLs wane through partial CE with Mn$^{2+}$ to produce crescent-shaped Cu$_{1.8}$S–MnS HNPLs as a result of anisotropic CE progression and the partial dissolution of Cu$_{1.8}$S. Control experiments indicated that both slow CE and halide-induced Cu$_{1.8}$S etching work in tandem to form crescent-shaped HNPLs. Then, the Cu$_{1.8}$S–MnS HNPLs wax upon the introduction of a quantity of Mn$^{2+}$ sufficient to replace all the Cu$^+$, resulting in the formation of complete-hexagonal MnS NPLs through the intraparticle S$^{2-}$ migration and growth of MnS on the previously missing regions. We have also shown that the waning and waxing strategy can be induced by other metal cations and can be applied to NRs and that the waxing cycle is a general and versatile process. These findings highlight an aspect of CE that allows us to modulate the morphologies of ionic NCs, thereby presenting a

methodology for the synthesis of structurally complex hetero-structured NCs with great potential for advanced optoelectronic and photocatalytic applications.

## Methods
### Chemicals
ODE (90%, Aldrich), OLAM (80%−90%, Aldrich), TOP (99%, Aldrich), tri-*n*-octylphosphine oxide (TOPO, ≥90%, Aldrich), *tert*-dodecanethiol (*t*-DDT; 98.5%, Aldrich), 1-dodecanethiol (DDT, ≥98%, Aldrich), TOA chloride (97%, Aldrich), CuCl$_2$ (99%, Aldrich), MnCl$_2$ (99%, Aldrich), ZnCl$_2$ (99.999%, Aldrich), CdCl$_2$ (99.999%, Aldrich), FeCl$_3$ (99.9%, Aldrich) and Cu(NO$_3$)$_2$·3H$_2$O (99%, TCI) were used as received. All solvents, such as hexane, isopropanol (IPA), and acetone, were of analytical grade and used as received.

### Synthesis of Cu$_{1.8}$S NPLs[22]
CuCl$_2$ (1.5 mmol), OLAM (2.46 mL), and ODE (15 mL) were mixed in a three-neck, round-bottom flask equipped with a Schlenk line. The system was vacuumed for 30 min at 80 °C and purged with N$_2$ before heating to 180 °C at a rate of 20 °C min$^{-1}$. *t*-DDT (6 mL) was rapidly injected into the flask at 120 °C. The reaction system was maintained at 180 °C for 5 min and then cooled to room temperature with a water bath. The reaction product was collected by centrifugation (9300× *g*) 2× with an acetone/IPA mixed solvent. The final black precipitate was redispersed in hexane for further use.

### Synthesis of Cu$_{1.8}$S NRs
Cu(NO$_3$)$_2$·3H$_2$O (2.33 mmol), TOPO (5.8 g), and ODE (30 mL) were combined in a three-neck, round-bottom flask equipped with a Schlenk line. The system was vacuumed for 30 min at 80 °C. The flask was purged with N$_2$ and then heated to 180 °C at a rate of 20

°C min$^{-1}$. When the reaction temperature reached 120 °C, a $t$-DDT:DDT mixture (volume ratio 10:1; 15 mL) was rapidly injected. The reaction system was maintained at 180 °C for 5 min and then cooled to room temperature with a water bath. The product NRs were collected by centrifugation (9300 × $g$) 2× with an acetone/IPA mixed solvent. The final black precipitate was resuspended in hexane for further use.

### CE reactions

The CE reactions of Cu$_{1.8}$S NPLs were carried out following a previously reported method[10]. Typically, MnCl$_2$ (0.4 mmol), OLAM (4 mL), and ODE (10 mL) were added to a three-neck, round-bottom flask equipped with a Schlenk line. The solution was heated to 100 °C under vacuum for 30 min and the flask was purged with N$_2$. The mixture was heated to 180 °C and maintained at this temperature for 30 min to form Mn–OLAM complexes, followed by cooling to 100 °C. In a separate vial, a mixture of Cu$_{1.8}$S NCs (containing 0.1 mmol S) and TOP (3 mL) was sonicated for 30 min and degassed under vacuum for 30 min. The Cu$_{1.8}$S NCs/TOP suspension was rapidly injected into the Mn–OLAM solution at 100 °C and the reaction proceeded for 5 min. Finally, the reaction solution was cooled to room temperature. The product was collected by centrifugation (9300 × $g$) 2× with a hexane/IPA mixed solvent and redispersed in hexane. The partial CE products, Cu$_{1.8}$S–MnS HNPLs, were synthesized by the same reaction procedure except that the quantity of MnCl$_2$ was changed ([MnCl$_2$]/[Cu$_{1.8}$S] = 0.25, 0.5, 0.75 or 1). CE reactions with Zn$^{2+}$ or Cd$^{2+}$ were carried out by the same method except that ZnCl$_2$ or CdCl$_2$ was used instead of MnCl$_2$. For the control experiment, a corresponding amount of TOA chloride was used instead of MnCl$_2$.

### Additional CE of Cu$_{1.8}$S–MnS HNPLs

A Cu$_{1.8}$S–MnS HNPLs solution was first prepared. Partial CE of Cu$_{1.8}$S NPLs was conducted using MnCl$_2$ ([MnCl$_2$]/[Cu$_{1.8}$S] = 0.5), and the crude Cu$_{1.8}$S–MnS HNPLs solution was kept at 100 °C. In a separate flask, a Mn–OLAM solution was prepared by mixing MnCl$_2$ (1.87 mmol), OLAM (4 mL), and ODE (10 mL) at 180 °C. The Mn-OLAM solution (3 mL) was rapidly injected into the crude Cu$_{1.8}$S–MnS HNPLs solution at 100 °C and the reaction proceeded for 5 min. The intermediate products were taken at various reaction times from 10 s to 5 min. The final product was collected by centrifuging (9300 × $g$) 2× with an acetone/IPA mixed solvent and resuspended in hexane. Additional CE of Cu$_{1.8}$S–MnS HNPLs with different metal cations were performed in the same way except that MnCl$_2$ was replaced by ZnCl$_2$, CdCl$_2$ or FeCl$_3$.

### Additional CE of Cu$_{1.8}$S–ZnS HNPLs

Additional CE of Cu$_{1.8}$S–ZnS HNPLs was conducted in the same way except that ZnCl$_2$ was used instead of MnCl$_2$ in the initial partial CE reaction.

### Sequential CE of Cu$_{1.8}$S NPLs or NRs with Mn$^{2+}$

MnCl$_2$ (0.1 mmol), OLAM (12 mL), and ODE (30 mL) were used for the 1$^{st}$ CE step. The mixture was kept at 180 °C for 30 min under N$_2$ to form Mn–OLAM complexes, followed by cooling to 100 °C. Then, a Cu$_{1.8}$S NCs/TOP suspension (9 mL, containing 1 mmol S) was rapidly injected into the Mn–OLAM solution at 100 °C. For the 2$^{nd}$ to 9$^{th}$ CE step, separately prepared Mn–OLAM solution (1 mL, containing 0.1 mmol MnCl$_2$) was injected to the remaining reaction solution. The reaction time for every CE step was 5 min. A portion of the reaction solution (1 mL) was taken after each CE step. CE products were collected by centrifuging (9300 × $g$) 2× with an acetone/IPA mixed solvent and resuspended in hexane.

### Calculation of Gibbs energy for the CE reaction

The $\Delta G_{CE}$ for CE reactions of Cu$_2$S with Mn$^{2+}$, Zn$^{2+}$, and Cd$^{2+}$ was calculated using standard Gibbs energy of formation ($\Delta G_f°$) and standard reduction potential ($E°$) based on a previous report[3].

$$Cu_2S + M^{2+} \rightarrow MS + 2Cu^+ \qquad (1)$$

$$Cu_2S \leftrightarrows 2Cu^0 + S^0 \qquad -\left(\Delta G_{f°}\right)_{Cu2S} \qquad (2)$$

$$2Cu^0 \leftrightarrows 2e^- + 2Cu^+ \qquad -2\left(E°\right)_{Cu+} \qquad (3)$$

$$M^{2+} + 2e^- \leftrightarrows M^0 \qquad \left(E°\right)_{M2+} \qquad (4)$$

$$M^0 + S^0 \leftrightarrows MS \qquad \left(\Delta G_{f°}\right)_{MS} \qquad (5)$$

$$\Delta G_{CE} = \left(\Delta G_{f°}\right)_{MS} - \left(\Delta G_{f°}\right)_{Cu2S} - 2F\left[\left(E°\right)_{M2+} - 2\left(E°\right)_{Cu+}\right] \qquad (6)$$

where F is Faraday's constant (9.6485 × 10$^4$ C mol$^{-1}$). The values of $\Delta G_f°$ and $E°$ were taken from the CRC handbook, as shown in Suppl. Table 1. The Gibbs free energies of CE reactions were calculated according to Eq. 6 and the results are summarized in Suppl. Table 1.

### Characterization

TEM observations were performed with a HT7820 (HITACHI) at an accelerating voltage of 100 kV. HRTEM, HAADF–STEM and STEM–EDX spectroscopy measurements were performed with a JEM-ARM200F (JEOL) at an accelerating voltage of 200 or 80 kV. The volumes of (H) NPLs were calculated using ImageJ software (ver. 1.54d). Macroscopic EDX measurements were carried out with an S-4800 (HITACHI). XRF spectroscopy was conducted with an EDX-7000 (SHIMADZU). XRD patterns were obtained with an X'Pert Pro MPD powder diffractometer (PANalytical) using CuKa radiation ($\lambda = 1.542$ Å) at 40 mA and 45 kV. UV–vis–NIR absorption spectra were obtained using a U-4100 spectrophotometer (HITACHI).

## Data availability

The data that support the findings of this study are available from the corresponding authors upon request.

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

## Acknowledgements

This research was supported by JST-CREST (Grant No. JPMJCR21B4) (T.T.), JST-FOREST (Grant No. JPMJFR213I) (M.S.), JSPS KAKENHI for Scientific Research S (Grant Nos. JP19H05634 and JP24H00053) (T.T.), Scientific Research B (Grant No. JP23H01802) (M.S.), Challenging Research (Exploratory) (Grant No. JP20K21236) (M.S.), JSPS Research Fellowship (Grant No. 19J23268) (Z.L.), Nagoya Institute of Technology molecule & material synthesis platform (Proposal No. JPMXP09S19NI0009 and JPMXP09S20NI0020) (M.S.) and Kyushu University advanced characterization platform (Proposal No. JPMXP09A21KU0380) (M. S.) as programs of "Nanotechnology Platform" and "Advanced Research Infrastructure for Materials and Nanotechnology in Japan (ARIM)". We thank Dr Jay Freeman at Edanz (https://jp.edanz.com/ac) for editing a draft of this manuscript.

## Author contributions

Z.L., M.S., and T.T. conceived the concept, designed the experiments, and wrote the paper. Z.L. synthesized and characterized the samples. T.A. performed HRTEM and STEM experiments.

## Competing interests

The authors declare no competing interests.
