## [Peer Review File · Nature Communications]

Waning-and-Waxing Shape Changes in Ionic Nanoplates upon Cation ExchangeEditorial Note: Parts of this Peer Review File have been redacted as indicated to remove third-party material where no permission to publish could be obtained.

Reviewer #1 (Remarks to the Author):

This manuscript provides a thorough and well-presented analysis of partial dissolution and re-structuring of $\text{Cu}_{1.8}\text{S}$ nanoplatelets during cation exchange with Mn to form MnS nanoplatelets. While similar observations of partial dissolution and re-structuring have been made in the literature, this is a more thorough study and more complete analysis than any others of which I am aware.

That being said, the authors should consider reports such as the following:

<https://doi.org/10.1021/acs.chemmater.1c01107>

<https://doi.org/10.1021/acsnano.8b01871>

<https://doi.org/10.1021/acs.chemmater.7b05198>

<https://doi.org/10.1021/jacs.8b03338>

<https://doi.org/10.1021/acsmaterialslett.0c00287>

to better put their work into context. They cite other papers by these same groups, but the five listed above may be more relevant than some of the ones they cite. These include examples of partial dissolution during cation exchange and similar formation of thin films of the cation exchange product on the surface of copper sulfide.

All of the composition measurements seem to be based on EDX and XRF. Because these are important to the manuscript's conclusions, it would be nice if at least a fraction of the EDX-based composition measurements could be benchmarked against a more quantitative and truly bulk method, such as ICP-OES or ICP-MS. The EDX measurements are only as good as the calibration of the instrument, and most of us use the standard calibration from the manufacturer, which may not match the local environment and can thus bias the samples. This is not essential, because the trends will clearly be the same, even if the quantification is slightly off.

Overall, despite some precedents in the literature, I believe that the clear and thorough observation and presentation of this waxing-waning phenomenon during cation exchange merits publication in a high impact journal.

Reviewer #2 (Remarks to the Author):

The manuscript demonstrates the waning and waxing-like shape transitions of $\text{Cu}_{1.8}\text{S}$ nanocrystals during the cation exchange process. The authors meticulously analyzed the mechanism behind these transitions, observing the crystallographic changes in the nanocrystals using techniques such as transmission electron microscopy and X-ray diffraction. These intriguing results challenge the conventional perspective that the collapse of the anion framework during the cation exchange process irreversibly prevents the recovery of the template nanocrystals' structure. These findings will pave the way for novel methodologies in cation exchange-driven nanocrystal synthesis. I believe the manuscript would be suitable for publication in *Nature Communications*, provided that some revisions are made, including responses to below comments.

1. In Fig. 1m, it is assumed that the atomic ratio of S should remain constant regardless of the $\text{ZnCl}_2/\text{Cu}_{1.8}\text{S}$ ratio, as the dissolved S during the waning process grows into the MnS shell. What could be the cause of the changes in the S ratio? Similarly, in Fig. 4e, f, g, what causes the S ratio to change according to the reaction time? I am curious about the extent of noise influence during the elemental quantification analysis using EDX, and I wonder how much difference there is compared to the results from the inductively coupled plasma measurement.
2. Based on the hard and soft acid and base (HSAB) theory, I believe that the decomposition of $\text{Cu}_{1.8}\text{S}$ could vary based on the base ligands used. If halides other than chloride are employed, or if the ratio of oleylamine to trioctylphosphine is adjusted, would it be possible to suppress the waning phenomenon even in the occurrence of a partial exchange?
3. There are errors in the figure numbers within the text. In the sentence on line 230, "Fig. 4" should

be corrected to "Fig. 5", and in the sentence on line 251, "Supplementary Fig. 9" should be corrected to "Supplementary Fig. 10".

Reviewer #3 (Remarks to the Author):

I co-reviewed this manuscript with one of the reviewers who provided the listed reports. This is part of the *Nature Communications* initiative to facilitate training in peer review and to provide appropriate recognition for Early Career Researchers who co-review manuscripts.

Reviewer #4 (Remarks to the Author):

Post-synthetic nanoscale cation exchange reactions have emerged as a powerful synthesis strategy to circumvent thermodynamic and kinetic limitations, providing pathways to nanocrystals and hetero-nanocrystals with compositions, morphologies, crystal structures, and hetero-architectures that would otherwise remain unavailable. These prospects have turned the study of nanoscale cation exchange reactions into a very relevant scientific endeavour, which has been attracting increasing interest in recent years. Cation exchange reactions are typically topotactic, thereby preserving the overall morphology and structure of the anionic sublattice of the parent nanocrystal into the product nanocrystal. There are however several recent examples of cation exchange reactions that are accompanied by morphological transformations.

In this contribution, Teranishi and coworkers report an intriguing and unprecedented example of severe morphological reorganization in response to a cation exchange reaction. The authors observe that upon partial aliovalent Mn^{2+} for Cu^{+} cation exchange, hexagonal roxbyite (rb) $Cu_{1.8}S$ nanoplates gradually transform into crescent-shaped $Cu_{1.8}S$ - MnS core-shell heteronanoplates, which upon further cation exchange evolve back into complete hexagonal nanoplates of wurtzite (wz) MnS . Similar cation exchange-induced transformations are also observed with Zn^{2+} , Cd^{2+} , and Fe^{3+} .

The work has been carried out very thoroughly and provides novel and significant insights that make it of high interest to a broad scientific community. Therefore, I think it merits publication in *Nature Communications*. However, there are several issues that should be addressed prior to its publication.

My main concern regards the mechanism proposed for the transformation, which is not clearly and convincingly presented in the paper. Moreover, the contextualization of the work with the most current literature in the field could be improved. I will address my concerns in more detail below.

1. On page 3 (Figure 1) the authors demonstrate that under excess Mn^{2+} the cation exchange reaction is clearly topotactic, maintaining the original shape and dimensions of the parent rb $Cu_{1.8}S$ nanoplates into the product wz MnS nanoplates (apart from a small expansion due to the slightly larger lattice volumes of wz- MnS). The authors imply in the proposed mechanism that also in this case the reaction proceeds through a 'Waning and Waxing' cycle. However, this is not necessarily the case, given that if the CE reaction is under kinetic control, it could proceed sufficiently fast to prevent the "waning" phase. Moreover, the data presented is insufficient to support this interpretation since intermediate samples were not collected under excess Mn^{2+} conditions, only under sub-stoichiometric conditions, which may favor waning for the reasons mentioned below.

2. The discussion concerning Figures 2 and 3 attributes the formation of the crescent-shaped $Cu_{1.8}S$ - MnS core-shell heteronanoplates to cation exchange. The authors suggest that Mn^{2+} for Cu^{+} cation exchange in the parent $Cu_{1.8}S$ NPLs begins from one side while S^{2-} is supplied from the opposite side through intraparticle migration. The dissolution of one half of the nanoplates is thus attributed to an imbalance between the rapid outward diffusion of Cu^{+} and the slower inward diffusion of Mn^{2+} .

However, this mechanism is inconsistent with the data presented because the MnS shell is observed to grow heteroepitaxially over one half of the parent Cu_{1.8}S NPLs, thereby making it thicker, while the other half gradually shrinks from the surface inwards. It has been shown for many different systems that core-shell heteronanocrystals formed by cation exchange reactions are characterized by superseded shell ingrowth, i.e., the total dimensions of the template nanocrystals are preserved while the shell grows thicker and the core shrinks due to the inwards diffusion flux of the guest cation and outward diffusion flux of the host cations (e.g., ZnSe-CdSe core-shell QDs [Groeneveld, E.; et al. ACS Nano (2013), 7, 7913], PbSe-CdSe core-shell QDs [Pietryga, J. M.; et al. J. Am. Chem. Soc. (2008), 130, 4879; Grodzinska, D.; et al., J. Mater. Chem. 2011, 21, 11556]. The observation reported in the present contribution is reminiscent of that recently reported by Xia, C. et al. [ACS Nano 15 (2021) 9987] concerning the formation mechanism of colloidal Janus-Type Cu_{2-x}S/CuInS₂ Heteronanorods via Seeded-Injection, which is shown to start with a partial aliovalent In³⁺ for Cu⁺ cation exchange in just one facet of the Cu_{2-x}S seed nanocrystals, converting it to wz-CuInS₂. As soon as this wz-CuInS₂ surface is formed, homoepitaxial growth of wz-CuInS₂ takes over, outcompeting the cation exchange reaction for the limited In³⁺ supply and gradually consuming the Cu_{2-x}S segment of the nanocrystals as sacrificial Cu⁺ sources. The formation of the Cu_{1.8}S-MnS core-shell heteronanoplates seems to proceed through a similar mechanism: Mn²⁺ for Cu⁺ cation exchange initially converts the bottom and top facets of the parent Cu_{1.8}S NPLs into wz-MnS, thus triggering a fast homoepitaxial overgrowth of a MnS shell using the available Mn²⁺ in solution and the other half of the NPLs as sacrificial sulfur sources. CE only resumes when the supply of S²⁻ has been exhausted (i.e., one half of the NPL has been entirely dissolved while the other is protected by the MnS shell) and proceeds inwards from the exposed side facets. The process would thus be triggered by a kinetic imbalance between the CE rates (too slow due to the low concentration of Mn²⁺) and the dissolution of the Cu_{1.8}S NPLs by Cl⁻ and homoepitaxial growth of MnS (both fast and kinetically coupled).

3. On page 10 the authors state that anisotropic CE progression and Cu_{1.8}S etching by Cl⁻ occur simultaneously to induce the transformation of hexagonal Cu_{1.8}S NPLs into crescent-shaped Cu_{1.8}S-MnS HNPLs. However, as discussed above this is inconsistent with the data presented and can be better explained by a kinetic imbalance.

4. On page 11 the authors invoke the nanoscale Kirkendall effect to explain the formation of the crescent-shaped Cu_{1.8}S-MnS HNPLs (i.e., the "waning" phase). However, this seems unlikely as the Kirkendall effect should rather result in the formation of hollow NPLs (i.e., dissolution of the center of the NPLs), as previously reported by Buhro and coworkers for CuInS₂ NCs (ref. [18]). To support this interpretation, the authors should also propose a reasonable explanation for the asymmetry observed in the process: why does one half of the NPL dissolve, while a MnS shell overgrows on the other? It seems more likely to me that the dissolution is driven by a kinetic imbalance in between etching rates by Cl⁻ and the CE rates (similarly to the formation of hollow Cu_{2-x}S nanocrystals through etching induced by GaCl₃ [Hinterding, et al.; ACS Nano (2019), 13, 12880]).

5. The authors convincingly demonstrate that chloride is responsible for the etching and dissolution observed during the waning phase. However, metal chlorides are widely used to deliver guest cations in cation exchange reactions and do not seem to induce any etching. Why do they act as etching agents under the conditions used in the present work? What is different with respect to the conditions typically used in other works?

6. The discussion on page 11 and elsewhere in the paper overlooks the importance of the solid-state diffusion rates of the guest cations in determining the overall heteroarchitecture of the product nanocrystals obtained by cation exchange reactions. This is a crucial point because it depends on the nature of both the guest cation and the host nanocrystal and on the temperature. Slower migration of the incoming guest cation does not necessarily result in collapse of the anionic sublattice and dissolution of the NC, but rather in the formation of heteronanocrystals with well-defined heterointerfaces, while fast diffusion results in homogeneous alloys. This has been extensively investigated by many groups, e.g., Schaak and coworkers [refs. 30-32], Swihart and coworkers

[Chem. Mater. (2018), 30, 1399], Robinson and coworkers [Nano Lett. (2014), 14, 7090], Donega and coworkers [Groeneveld, E.; et al. ACS Nano (2013), 7, 7913; Hinterding, et al.; ACS Nano (2019), 13, 12880], Manna and coworkers [ref. 4].

7. Page 12, line 230: "Fig. 4" should be "Fig. 5".

8. The mechanism proposed for the "waxing" process is rather vague and does not seem to be fully supported by the data presented. The cartoon in Figure 5 assumes that the starting configuration for the waxing process corresponds to the end configuration of the waning process. This is however not supported by the TEM images and STEM-EDX maps presented in Figure 5, which are consistent with the intermediate configuration shown in the cartoon in panel I of figure 5 but provide no evidence for the starting configuration. Moreover, the mechanism proposed leaves many intriguing questions without satisfactory answers, e.g.: (a) why would the MnS shell formed during the waning phase suddenly redissolve and regrow to regenerate the missing half of the nanoplate? The conditions are clearly favorable to MnS growth, so why would they at the same time promote the dissolution of what was seemingly a very stable MnS shell? (b) And why would the thickness of the NPL revert back to the original thickness? What prevents the CE to proceed and convert the Cu_{1.8}S core into MnS while additional MnS grow on the sides of NPLs concomitantly reconstructs the missing half of the nanoplate, yielding a thicker hexagonal NPL? (c) Why does the Cu_{1.8}S that was stored in the core of the core-shell Cu_{1.8}S-MnS heteroNPLs suddenly migrates to the surface and spreads itself over just one of the wide facets? The authors are either missing several crucial intermediate steps in the conversion of the left cartoon to the middle cartoon or are mistaken in their mechanism proposed for the waxing process.

9. In the final section of the paper the authors discuss their experiments with nanorods. How do their observations differ from what has already been extensively reported by Schaak and coworkers? (e.g., refs. [30-32])

10. Claim of universality at the end of paper is likely exaggerated. The authors investigated four different cations: Mn²⁺, Zn²⁺, Cd²⁺ and Fe³⁺. The waning-waxing cycle process occurs only for Mn²⁺ and Zn²⁺. Cd²⁺ gives Janus-type hetero-NPLs and Fe³⁺ gives homogeneous CuFeS₂. What determines the different outcomes? Likely the different solid-state diffusion rates of the different cations, but this is an aspect that is largely overlooked in the mechanism proposed in the paper.

Manuscript ID: NCOMMS-23-46418-T

Title: 'Waning and Waxing' Shape Change in Ionic Nanocrystals upon Cation Exchange

Authors: Zhanzhao Li, Masaki Saruyama, Toru Asaka and Toshiharu Teranishi

We outlined our replies (in blue) and manuscript revisions (in red) in the text below. Changes in the resubmitted manuscript are marked in yellow for reviewers only.

Author replies to the comments of Reviewer 1

This manuscript provides a thorough and well-presented analysis of partial dissolution and re-structuring of Cu_{1.8}S nanoplatelets during cation exchange with Mn to form MnS nanoplatelets. While similar observations of partial dissolution and re-structuring have been made in the literature, this is a more thorough study and more complete analysis than any others of which I am aware.

Author reply: We are grateful for your positive comments and constructive suggestions to improve our manuscript.

Q1: That being said, the authors should consider reports such as the following:

<https://doi.org/10.1021/acs.chemmater.1c01107>

<https://doi.org/10.1021/acsnano.8b01871>

<https://doi.org/10.1021/acs.chemmater.7b05198>

<https://doi.org/10.1021/jacs.8b03338>

<https://doi.org/10.1021/acsmaterialslett.0c00287>

to better put their work into context. They cite other papers by these same groups, but the five listed above may be more relevant than some of the ones they cite. These include examples of partial dissolution during cation exchange and similar formation of thin films of the cation exchange product on the surface of copper sulfide.

Author reply: Thank you for your suggestions. We additionally cited 4 out of 5 suggested papers to the text. The first one [<https://doi.org/10.1021/acs.chemmater.1c01107>] shows the shell formation on the host nanorod via anion exchange reaction ($\text{Cu}_{2-x}\text{S} \rightarrow \text{Cu}_{2-x}\text{Te}$). This is a relevant example, but the kinetics of anion exchange is quite different from that of cation exchange. The anion exchange reaction proceeds quite slow (slow ion diffusion) and the crystal structure easily changed due to the large ion radius. Therefore, we decided not to refer to the first paper. However, other 4 papers should be cited as relevant examples for anisotropic cation exchange progress (e.g.,

formation of Janus-like structure or multi-patchy structure) and etching of $\text{Cu}_{1.8}\text{S}$ region with TOP and O_2 .

Manuscript revision:

In the revised manuscript, we additionally cited the above 4 papers as refs 26, 27, 28 and 29, respectively.

Q2: All of the composition measurements seem to be based on EDX and XRF. Because these are important to the manuscript's conclusions, it would be nice if at least a fraction of the EDX-based composition measurements could be benchmarked against a more quantitative and truly bulk method, such as ICP-OES or ICP-MS. The EDX measurements are only as good as the calibration of the instrument, and most of us use the standard calibration from the manufacturer, which may not match the local environment and can thus bias the samples. This is not essential, because the trends will clearly be the same, even if the quantification is slightly off.

Author reply: Thank you for your suggestion. First, we tried ICP-OES analysis of the cation exchanged products, but the decomposition of metal sulfide nanocrystals by strong acid treatment during ICP sample preparation led to the volatilization of sulfur as H_2S gas from the solution, which results in a lower estimation of S. To assess the accuracy of EDX measurements in determining Cu:Mn:S mole ratios, we prepared the mixed aqueous solutions from standard aqueous solutions for ICP analysis containing fixed concentration of Cu, Mn, or S (1000 ± 10 mg/L for Cu and Mn, 1000 ± 20 mg/L for S, purchased from FUJIFILM Wako). The mixed solutions were dried to obtain the solids containing Cu, Mn, and S for EDX measurements. Three distinct samples were prepared with different Cu:Mn:S ratios, so as to replicate samples with different degrees of cation exchange (CE) progression (Table R1). The results revealed that the mole ratios estimated from EDX analysis were in good agreement with the calculated values (Fig. R1). The Cu/Mn mole ratios also closely match the theoretical values, indicating the suitability of EDX analysis in estimating the degree of CE progression of $\text{Cu}_{1.8}\text{S}$ with Mn^{2+} . Consequently, we would like to assert that EDX measurements give accurate composition values of cation-exchanged products in our experiments.

Table R1. Amounts of ICP standard solutions for sample preparation, calculated mole ratios, and mole ratios estimated by EDX measurements. Five distinct areas were measured for EDX analysis to obtain the average values and standard deviations.

		Cu	Mn	S
Sample 1	ICP standard solution (μL)	200	500	300
	Calculated mole ratio (%)	14.6	42.1	43.3
	Mole ratio from EDX (%)	15.1 ± 1.3	42.3 ± 0.8	42.6 ± 1.3
Sample 2	ICP standard solution (μL)	350	350	300
	Calculated mole ratio (%)	25.9	30.0	44.1
	Mole ratio from EDX (%)	25.7 ± 1.4	27.1 ± 3.6	47.2 ± 4.2
Sample 3	ICP standard solution (μL)	550	150	300
	Calculated mole ratio (%)	41.7	13.2	45.1
	Mole ratio from EDX (%)	38.6 ± 5.0	12.0 ± 1.1	49.5 ± 4.9

Figure R1. EDX results of three solids containing Cu, Mn, and S with controlled mole ratios. Gray and bars stand for calculated and experimental EDX values, respectively. Five distinct areas were measured for EDX analysis to obtain the average values and standard deviations.

Overall, despite some precedents in the literature, I believe that the clear and thorough observation and presentation of this waxing-waning phenomenon during cation exchange merits publication in a high impact journal.

We appreciate your positive recommendation.

Author replies to the comments of Reviewer 2

The manuscript demonstrates the waning and waxing-like shape transitions of $\text{Cu}_{1.8}\text{S}$ nanocrystals during the cation exchange process. The authors meticulously analyzed the mechanism behind these transitions, observing the crystallographic changes in the nanocrystals using techniques such as transmission electron microscopy and X-ray diffraction. These intriguing results challenge the conventional perspective that the collapse of the anion framework during the cation exchange process irreversibly prevents the recovery of the template nanocrystals' structure. These findings will pave the way for novel methodologies in cation exchange-driven nanocrystal synthesis. I believe the manuscript would be suitable for publication in Nature Communications, provided that some revisions are made, including responses to below comments.

Author reply: We are grateful for your positive comments and constructive suggestions to improve our manuscript.

Q1: In Fig. 1m, it is assumed that the atomic ratio of S should remain constant regardless of the $\text{MnCl}_2/\text{Cu}_{1.8}\text{S}$ ratio, as the dissolved S during the waning process grows into the MnS shell. What could be the cause of the changes in the S ratio? Similarly, in Fig. 4e, f, g, what causes the S ratio to change according to the reaction time? I am curious about the extent of noise influence during the elemental quantification analysis using EDX, and I wonder how much difference there is compared to the results from the inductively coupled plasma measurement.

Author reply: Thank you for your comments. According to the chemical formula, the mole ratios of S in $\text{Cu}_{1.8}\text{S}$ and in MS (M = Mn, Zn, Cd) should be 36% and 50%, respectively. Therefore, the mole ratio of S in the product should increase from 36% to 50% as the M^{2+} cation exchange (CE) of $\text{Cu}_{1.8}\text{S}$ proceeds. This trend is consistent with our EDX results in Fig. 1m (and other EDX results). The change in the mole ratios of S in Fig. 4e–g can be explained by the same story, while the mole ratios of S plateau out at <50% due to the formation of $\text{Cu}_{1.8}\text{S}$ –MS via partial CE reactions with insufficient M^{2+} cations.

Figure R2 shows representative raw EDX spectra of $\text{Cu}_{1.8}\text{S}$ nanoplates (NPLs) before and after CE reactions. Long accumulation time gave smooth spectrum and high peak/background intensity ratio, allowing to calculate each mole ratio with small uncertainty. The difference in mole ratios with and without background subtraction is only approximately 0.5% for each element, also indicating the strong peak intensity (Table R2) (we use background-subtracted value in the manuscript).

Figure R2. Representative raw EDX spectra of $\text{Cu}_{1.8}\text{S}$ NPLs, $\text{Cu}_{1.8}\text{S}$ –MnS heterostructured NPLs (HNPLs) and MnS NPLs. Green regions show background intensity. Signals from C, O and Al were detected from carbon tape substrates (C and O) and sample stages (Al). Cu-K, Mn-K and S-K lines are used to quantify mole ratios.

Table R2. Comparison of mole ratios calculated from EDX spectra in Fig. R2 with and without background subtraction.

		mol%		
		Cu	Mn	S
$\text{Cu}_{1.8}\text{S}$ NPLs	Raw data	64.5	-	35.5
	Background subtracted data	65.0	-	35.0
$\text{Cu}_{1.8}\text{S}$ -MnS HNPLs	Raw data	34.2	23.6	42.2
	Background subtracted data	34.3	23.5	42.2
MnS NPLs	Raw data	4.4	48.6	47.0
	Background subtracted data	3.7	49.2	47.1

Then, we investigated the difference between the EDX and ICP results. First, we tried ICP-OES analysis of the cation exchanged products, but the decomposition of metal sulfide nanocrystals by strong acid treatment during ICP sample preparation led to the volatilization of sulfur as H_2S gas from the solution, which results in a lower estimation of S. To assess the accuracy of EDX measurements in determining Cu:Mn:S mole ratios, we prepared the mixed aqueous solutions from standard aqueous solutions for ICP analysis containing fixed concentration of Cu, Mn, or S (1000 ± 10 mg/L for Cu and Mn, 1000 ± 20 mg/L for S, purchased from FUJIFILM Wako). The mixed solutions were dried to obtain the solids containing Cu, Mn, and S for EDX measurements. Three distinct samples were prepared with different Cu:Mn:S ratios, so as to replicate samples with different degrees of cation exchange (CE) progression (Table R3). The results revealed that the

mole ratios estimated from EDX analysis were in good agreement with the calculated values (Fig. R3). The Cu/Mn mole ratios also closely match the theoretical values, indicating the suitability of EDX analysis in estimating the degree of CE progression of $\text{Cu}_{1.8}\text{S}$ with Mn^{2+} . Consequently, we would like to assert that EDX measurements give accurate composition values of cation-exchanged products in our experiments.

Table R3. Amounts of ICP standard solutions for sample preparation, calculated mole ratios, and mole ratios estimated by EDX measurements. Five distinct areas were measured for EDX analysis to obtain the average values and standard deviations.

		Cu	Mn	S
Sample 1	ICP standard solution (μL)	200	500	300
	Calculated mole ratio (%)	14.6	42.1	43.3
	Mole ratio from EDX (%)	15.1 ± 1.3	42.3 ± 0.8	42.6 ± 1.3
Sample 2	ICP standard solution (μL)	350	350	300
	Calculated mole ratio (%)	25.9	30.0	44.1
	Mole ratio from EDX (%)	25.7 ± 1.4	27.1 ± 3.6	47.2 ± 4.2
Sample 3	ICP standard solution (μL)	550	150	300
	Calculated mole ratio (%)	41.7	13.2	45.1
	Mole ratio from EDX (%)	38.6 ± 5.0	12.0 ± 1.1	49.5 ± 4.9

Figure R3. EDX results of three solids containing Cu, Mn, and S with controlled mole ratios. Gray and bars stand for calculated and experimental EDX values, respectively. Five distinct areas were measured for EDX analysis to obtain the average values and standard deviations.

Q2: Based on the hard and soft acid and base (HSAB) theory, I believe that the decomposition of $\text{Cu}_{1.8}\text{S}$ could vary based on the base ligands used. If halides other than chloride are employed, or if the ratio of oleylamine to trioctylphosphine is adjusted, would it be possible to suppress the waning phenomenon even in the occurrence of a partial exchange?

Author reply: Thank you for your valuable suggestion. We used other Mn halides, MnBr_2 and MnI_2 , as Mn precursors for CE reactions. MnF_2 was not used because MnF_2 was not dissolved in oleylamine/octadecene solvent. For both cases, the waning and waxing phenomenon took place, as observed in the case using MnCl_2 (Fig. R4). From the viewpoint of the HSAB theory, all halides used here can be regarded as soft bases that have strong coordination ability with soft Cu^+ to cause etching of $\text{Cu}_{1.8}\text{S}$ (Table R4). Therefore, similar products were obtained regardless of halide anions.

Figure R4. TEM images of CE products using MnBr_2 [$\text{MnBr}_2/\text{Cu}_{1.8}\text{S}$ = (a) 0.5 mol/mol and (b) 1 mol/mol] and MnI_2 [(c) $\text{MnI}_2/\text{Cu}_{1.8}\text{S}$ = 0.5 mol/mol and (d) 1 mol/mol].

Table R4. The hardness of halide anions, trioctylphosphine (TOP), and oleylamine as Lewis bases, and Cu⁺ and Mn²⁺ as Lewis acids. (Refs: Pearson et al., *Inorg. Chem.* 1988, 27, 734–740; Alivisatos et al., *J. Phys. Chem. C* 2013, 117, 19759–19770.)

		Hardness (eV)
Lewis bases	Cl ⁻	4.7
	Br ⁻	4.2
	I ⁻	3.7
	TOP	6.0
	Oleylamine	8.0
Lewis acids	Cu ⁺	6.3
	Mn ²⁺	9.0

Then, we conducted experiments employing various TOP concentrations. Typically, we used 3 mL of TOP for injection to synthesize Cu_{1.8}S NPLs. The TOP quantity was varied by diluting the injection solution with octadecene, while maintaining the total volume at 3 mL (Table R5). The Mn content in the resulting products decreased with decreasing the TOP quantity. When TOP was reduced to 0.3 mL or 1 mL (1/10 or 1/3 of the typical amount, respectively) with 0.5 equivalent of MnCl₂ to Cu_{1.8}S, the Mn/Cu mole ratio in the products was 0.08 or 0.17, respectively (Entries 1 and 2 in Table R5). These ratios are notably lower than that when using 3 mL of TOP (0.69, Entry 4 in Table R5), which is in good agreement with the hypothesis that TOP induces CE by extracting Cu⁺ from the host Cu_{1.8}S NCs.

In comparison with the case using 3 mL of TOP, the morphological transformation was less obvious even with a similar Mn/Cu mole ratio in the products (Fig. R5). For instance, when using 3 mL of TOP, one sides of the NPLs initiated dissolution at a Mn/Cu mole ratio of 0.06 (Fig. R5d), while no evident dissolution was observed at a Mn/Cu mole ratio of 0.08 in the case using 0.3 mL of TOP (Fig. R5a). In the case using 1 mL of TOP, one sides of the NPLs slightly dissolved at a Mn/Cu mole ratio of 0.17 (Fig. R5b), while half of the NPLs underwent severe decomposition even at a Mn/Cu mole ratio of 0.13 using 3 mL of TOP (Fig. R5e). These different degrees of transformation can be explained by the role of TOP as an etching reagent, as reported in the literature, indicating that TOP dissolves Cu_{2-x}S NCs by forming phosphine sulfide species [A. Nelson et al., *Chem. Mater.* 2016, 28, 8530. (ref. 35)]. In our case, not only halide anions but also TOP quantity promoted the transformation of NPLs by hastening the dissolution of the Cu_{1.8}S NPLs, resulting in a less degree of transformation when using smaller TOP quantities.

Nevertheless, a subtle anisotropic change in the shape of NPLs could be observed when using 1 mL of TOP (as indicated by arrows in Fig. R5b), suggesting that the waning process initiates even at a lower TOP concentration. To observe further transformation with 1 mL of TOP, a larger quantity of MnCl₂ was employed to enhance Mn²⁺ CE (MnCl₂/Cu_{1.8}S = 1, Entry 3 in Table R5). As expected, the Mn/Cu mole ratio in the CE product increased to 0.29, and one sides of the NPLs underwent severe dissolution (Fig. R5c). These results strongly indicate that the anisotropic CE proceeds to give anisotropic shapes, even when the extracted quantity of Cu⁺ is reduced by decreasing the TOP quantity. It is also suggested that, even at low CE ratios, shape changes may occur due to the intraparticle migration of cations if there is a significant difference in cation diffusion rates (Mn²⁺ vs. Cu⁺).

In summary, even when different halide salts and smaller TOP quantities were employed, the anisotropic shape changes proceeded, following the waning process. This is consistent with the explanation that the shape changes arise from differences in cation diffusion rates during CE, suggesting that this phenomenon is unique to the Cu⁺ and Mn²⁺ combination.

Table R5. Reaction conditions and EDX results of the products obtained by using different TOP quantities in CE. *Entry 4 corresponds to the result in the manuscript (Fig. 1m).

Entry	Injection solvent (mL)		MnCl ₂ /Cu _{1.8} S (mol/mol)	EDX results (mol%)			
	TOP	octadecene		Cu	Mn	S	Mn/Cu
1	0.3	2.7	0.5	61.3	4.8	33.9	0.08
2	1	2	0.5	55.9	9.6	34.5	0.17
3	1	2	1	51.4	14.8	33.8	0.29
4*	3	0	0.5	34.0	23.5	42.5	0.69

Figure R5. TEM images of product NPLs: (a–c) products of entry 1–3 listed in Table R5; (d–e) products shown in Extended Fig. 1a in the manuscript. Mn/Cu values represent the Mn to Cu mole ratio from EDX measurements. Scale bars are 100 nm. Arrows in (b) indicate anisotropic dissolution of NPLs.

Q3: There are errors in the figure numbers within the text. In the sentence on line 230, “Fig. 4” should be corrected to “Fig. 5”, and in the sentence on line 251, “Supplementary Fig. 9” should be corrected to “Supplementary Fig. 10”.

Author reply: Thank you for your careful reading.

Manuscript revision:

We revised the Figure numbers on pages 13 and 14.

Author replies to the comments of Reviewer 3

Author reply: Thank you for reviewing our manuscript. We would like you to review our revised manuscript.

Author replies to the comments of Reviewer 4

Post-synthetic nanoscale cation exchange reactions have emerged as a powerful synthesis strategy to circumvent thermodynamic and kinetic limitations, providing pathways to nanocrystals and hetero-nanocrystals with compositions, morphologies, crystal structures, and hetero-architectures that would otherwise remain unavailable. These prospects have turned the study of nanoscale cation exchange reactions into a very relevant scientific endeavour, which has been attracting increasing interest in recent years. Cation exchange reactions are typically topotactic, thereby preserving the overall morphology and structure of the anionic sublattice of the parent nanocrystal into the product nanocrystal. There are however several recent examples of cation exchange reactions that are accompanied by morphological transformations.

In this contribution, Teranishi and coworkers report an intriguing and unprecedented example of severe morphological reorganization in response to a cation exchange reaction. The authors observe that upon partial aliovalent Mn^{2+} for Cu^+ cation exchange, hexagonal roxbyite (rb) $\text{Cu}_{1.8}\text{S}$ nanoplates gradually transform into crescent-shaped $\text{Cu}_{1.8}\text{S}$ - MnS core-shell heteronanoplates, which upon further cation exchange evolve back into complete hexagonal nanoplates of wurtzite (wz) MnS . Similar cation exchange-induced transformations are also observed with Zn^{2+} , Cd^{2+} , and Fe^{3+} .

The work has been carried out very thoroughly and provides novel and significant insights that make it of high interest to a broad scientific community. Therefore, I think it merits publication in Nature Communications. However, there are several issues that should be addressed prior to its publication.

My main concern regards the mechanism proposed for the transformation, which is not clearly and convincingly presented in the paper. Moreover, the contextualization of the work with the most current literature in the field could be improved. I will address my concerns in more detail below.

Author reply: We are grateful for your positive comments and constructive suggestions. Your numerous valuable suggestions encouraged us to reconsider the transformation mechanism during cation exchange (CE). Based on your comments, we proposed more detailed discussions.

Q1: On page 3 (Figure 1) the authors demonstrate that under excess Mn^{2+} the cation exchange reaction is clearly topotactic, maintaining the original shape and dimensions of the parent rb $\text{Cu}_{1.8}\text{S}$ nanoplates into the product wz MnS nanoplates (apart from a small expansion due to the slightly larger lattice volumes of wz- MnS). The authors imply in the proposed mechanism that also in this case the reaction cation exchange proceeds through a ‘Waning and Waxing’ cycle. However, this is not necessarily the case, given that if the CE reaction is under kinetic control, it could proceed

sufficiently fast to prevent the “waning” phase. Moreover, the data presented is insufficient to support this interpretation since intermediate samples were not collected under excess Mn^{2+} conditions, only under sub-stoichiometric conditions, which may favor waning for the reasons mentioned below.

Author reply: As you pointed out, Figure 1 contains the results obtained by using excess Mn^{2+} . To clarify whether the CE proceeds through a ‘Waning and Waxing’ cycle at the ratio of $\text{MnCl}_2/\text{Cu}_{1.8}\text{S} \geq 1$ mol/mol, intermediate products were characterized by TEM and XRD. Since the CE reaction proceeded too rapidly to obtain intermediate products at $\text{MnCl}_2/\text{Cu}_{1.8}\text{S} = 4$ mol/mol, we employed the reaction conditions of $\text{MnCl}_2/\text{Cu}_{1.8}\text{S} = 1$ mol/mol (Fig. R6). Within the 5 min reaction, a clear ‘Waning and Waxing’ cycle was observed. Consequently, we would like to claim that the CE proceeds through a ‘Waning and Waxing’ cycle even under conditions of elevated Mn^{2+} concentrations.

Figure R6. (a–f) TEM images and (g) XRD patterns of the products formed during CE of $\text{Cu}_{1.8}\text{S}$ NPLs with Mn^{2+} (reaction conditions: $\text{MnCl}_2/\text{Cu}_{1.8}\text{S} = 1$ mol/mol) at (a) 0 s, (b) 10 s, (c) 30 s, (d) 1 min, (e) 3 min and (f) 5 min.

Q2: The discussion concerning Figures 2 and 3 attributes the formation of the crescent-shaped $\text{Cu}_{1.8}\text{S}\text{-MnS}$ core-shell heteronanoplates to cation exchange. The authors suggest that Mn^{2+} for Cu^+ cation exchange in the parent $\text{Cu}_{1.8}\text{S}$ NPLs begins from one side while S^{2-} is supplied from the opposite side through intraparticle migration. The dissolution of one half of the nanoplates is thus attributed to an imbalance between the rapid outward diffusion of Cu^+ and the slower inward diffusion of Mn^{2+} . However, this mechanism is inconsistent with the data presented because the MnS shell is observed to grow heteroepitaxially over one half of the parent $\text{Cu}_{1.8}\text{S}$ NPLs, thereby making it thicker, while the other half gradually shrinks from the surface inwards. It has been shown

for many different systems that core-shell heteronanocrystals formed by cation exchange reactions are characterized by superseded shell ingrowth, i.e., the total dimensions of the template nanocrystals are preserved while the shell grows thicker and the core shrinks due to the inwards diffusion flux of the guest cation and outward diffusion flux of the host cations (e.g., ZnSe-CdSe core-shell QDs [Groeneveld, E.; et al. ACS Nano (2013), 7, 7913], PbSe-CdSe core-shell QDs [Pietryga, J. M.; et al. J. Am. Chem. Soc. (2008), 130, 4879; Grodzinska, D.; et al., J. Mater. Chem. 2011, 21, 11556]. The observation reported in the present contribution is reminiscent of that recently reported by Xia, C. et al. [ACS Nano 15 (2021) 9987] concerning the formation mechanism of colloidal Janus-Type $\text{Cu}_{2-x}\text{S}/\text{CuInS}_2$ Heteronanorods via Seeded-Injection, which is shown to start with a partial aliovalent In^{3+} for Cu^+ cation exchange in just one facet of the Cu_{2-x}S seed nanocrystals, converting it to wz-CuInS₂. As soon as this wz-CuInS₂ surface is formed, homoepitaxial growth of wz-CuInS₂ takes over, outcompeting the cation exchange reaction for the limited In^{3+} supply and gradually consuming the Cu_{2-x}S segment of the nanocrystals as sacrificial Cu^+ sources. The formation of the $\text{Cu}_{1.8}\text{S}-\text{MnS}$ core-shell heteronanoplates seems to proceed through a similar mechanism: Mn^{2+} for Cu^+ cation exchange initially converts the bottom and top facets of the parent $\text{Cu}_{1.8}\text{S}$ NPLs into wz-MnS, thus triggering a fast homoepitaxial overgrowth of a MnS shell using the available Mn^{2+} in solution and the other half of the NPLs as sacrificial sulfur sources. CE only resumes when the supply of S^{2-} has been exhausted (i.e., one half of the NPL has been entirely dissolved while the other is protected by the MnS shell) and proceeds inwards from the exposed side facets. The process would thus be triggered by a kinetic imbalance between the CE rates (too slow due to the low concentration of Mn^{2+}) and the dissolution of the $\text{Cu}_{1.8}\text{S}$ NPLs by Cl^- and homoepitaxial growth of MnS (both fast and kinetically coupled).

Author reply: Thank you for your valuable suggestion. Firstly, no discussion regarding the relative rates of outward/inward cation diffusion was made in the above works on core-shell formation through CE reactions that you suggested. In the case of core-shell heteronanocrystals, since no extraction reagents such as TOP (as Cu^+ extractor) were employed, the slow outward diffusion of host cations might prevent deformation of host nanocrystals (NCs) due to the similar relative inward diffusion rate of guest cations. Additionally, shape changes are not clearly observed for small spherical NCs. In contrast, the shape changes can be clearly observed for larger plates, indicating the obvious Kirkendall-type shape transformations in several cases (e.g., Fig. R7). The conventional Kirkendall effect typically results in the formation of hollow structures for nanospheres (isotropic transformation) and rings or biconcave shapes for NPLs (isotropic transformation in in-plane direction). The Kirkendall effect should also take place at heterointerfaces of anisotropic structures due to the imbalance in atomic diffusion to achieve the anisotropic-to-anisotropic shape transformation. We believe that anisotropic shape changes

induced by a significant difference in atomic diffusion rates of host and guest cations in the in-plane direction are still within the scope of the Kirkendall effect.

Figure R7. Deformation of NPLs caused by large differences in relative outward/inward cation diffusion rates. (Reprinted with permission from *Chem. Mater.* **2019**, 31, 1990 and *J. Am. Chem. Soc.* **2011**, 133, 14500. Copyright 2019 and 2011 American Chemical Society.)

However, as you suggested, we should consider the MnS-shell deposition mechanism even during the waning stage. After the $\text{Cu}_{1.8}\text{S}$ NPLs are partially etched by Cl^- and/or TOP, the dissolved S^{2-} can react with Mn^{2+} , causing the growth of MnS on residual $\text{Cu}_{1.8}\text{S}$. Nevertheless, the MnS deposition process cannot explain the shell formation only at limited positions of the $\text{Cu}_{1.8}\text{S}$ NPLs. Moreover, if the deposition of metal sulfides after $\text{Cu}_{1.8}\text{S}$ etching were dominant, it would be quite difficult to preserve the plate morphology during partial Cd^{2+} CE with chloride precursor under the same temperature and time conditions, as in the cases of Mn^{2+} and Zn^{2+} .

Hence, based on the previous examples such as $\text{Cu}_2\text{S}/\text{CuInS}_2$ [C. Xia et al. *ACS Nano* 2021, 15, 9987. (ref. 36)], it is more conceivable that an initially cation exchanged position serves as the starting point for subsequent processes (further CE and MnS-shell deposition) (Fig. R8). Relatively slow inward Mn^{2+} diffusion causes interparticle diffusion of ions to create structural defects on the opposite side of NPL, providing exposed fresh and unstable surface (Fig. R8b). This exposed area becomes the starting point for accelerated $\text{Cu}_{1.8}\text{S}$ etching by Cl^- and/or TOP to release S^{2-} into the solution, leading to the deposition of MnS from the dissolved S^{2-} and Mn^{2+} (Fig. R8c). Similar phenomena have been observed in the formation of hollow and pits through etching processes with Cl^- , TOP and/or O_2 [Y. Xiong et al., *Angew. Chem. Int. Ed.* **2005**, 44, 7913; A. Nelson et al., *Chem. Mater.* **2016**, 28, 8530. (ref. 35)]. It is also believed that CE continues during the MnS deposition process. This is inferred from the position of $\text{Cu}_{1.8}\text{S}$ part in the crescent-shaped NPL, not near the edge but towards the center (Fig. R8d).

Figure R8. Schematic of the formation of crescent-shaped $\text{Cu}_{1.8}\text{S}/\text{MnS}$ heterostructured NPL (HNPL) including CE and MnS deposition mechanisms from the side view.

Manuscript revision:

Based on the discussion above, we revised the manuscript and added Fig. R8 as Supplementary Fig. 7 to add the MnS shell deposition processes to the transformation mechanism during waning process.

Page 10, 3rd paragraph:

Based on the above characterization, we speculate the transformation mechanism of the waning process. The formation of NPLs with non-uniform thickness in the early stage (e.g., ~ 3 nm and ~ 9 nm at 10 s) from the flat $\text{Cu}_{1.8}\text{S}$ NPLs (5.3 nm) is likely initiated by anisotropic intraparticle ion migration within individual NPLs after the CE with Mn^{2+} from one side of $\text{Cu}_{1.8}\text{S}$ NPLs^{15,16,25}. Initiation of CE from a single location on a NC forms the starting point for the subsequent anisotropic CE, often leading to the formation of Janus-type heterostructure, as observed in many cases^{8,26–34}. Explanations for this phenomenon have been often provided by the formation of a crystallographically stable heterointerface^{31,33} and/or the presence of a high activation energy for the CE reaction^{30,34}. These explanations would also apply to our case, where CE of the $\text{Cu}_{1.8}\text{S}$ NPL with Mn^{2+} started from a single location. Subsequently, the imbalance between the rapid outward diffusion of the host Cu^+ and the slow inward diffusion of the guest Mn^{2+} (as shown later) causes the anisotropic shape transformations¹⁶. Such a transformation triggered by a large difference in inward/outward cation migration rates has been shown in several cases, which are often explained as nanoscale Kirkendall effect^{15–18}. In the case of NPLs, unique ring^{15,18} and biconcave-shaped¹⁶ nanostructure have been obtained through intraparticle ion migration in in-plane direction during CE initiated from all edges of the NPLs. In our case, the progress of CE with Mn^{2+} from one side

of $\text{Cu}_{1.8}\text{S}$ NPLs causes the directional in-plane ion migration, leading to the formation of anisotropic NPLs with non-uniform thickness. The CE continuously propagates MnS phases in NPLs from the edge, which is evidenced by the position of $\text{Cu}_{1.8}\text{S}$ phases within the crescent shaped NPLs, not near the edge but towards the centre (Fig. 2).

In addition to the anisotropic Kirkendall-type intraparticle ion migration, the decomposition of $\text{Cu}_{1.8}\text{S}$ NPLs triggered by strong coordination between Cu^+ and Cl^- promotes the large deformation. Once partial CE with Mn^{2+} occurs at a single location of NPL, interparticle diffusion of ions creates structural defects on the opposite side of NPL to provide exposed fresh and unstable $\text{Cu}_{1.8}\text{S}$ surface as the starting point for accelerated etching by Cl^- (Supplementary Fig. 7). On such a highly reactive surface, TOP is also expected to act as a supplemental etching agent for S^{2-} , further accelerating NPLs dissolution³⁵. The exposed $\text{Cu}_{1.8}\text{S}$ not covered by a MnS shell in the intermediate (observed at 10 s) is susceptible to etching and subsequently disappears (after 1 min, Fig. 3c).

Another plausible reaction in the waning process is the MnS deposition on NPLs. After the $\text{Cu}_{1.8}\text{S}$ NPLs is partially etched, the dissolved S^{2-} reacts with Mn^{2+} to cause the growth of MnS on residual $\text{Cu}_{1.8}\text{S}$ (as shown later)³⁶. Considering that the CE generally proceeds from an edge of $\text{Cu}_{1.8}\text{S}$ NPLs, the MnS shells on both faces of $\text{Cu}_{1.8}\text{S}$ NPLs might grow through this MnS deposition mechanism (Supplementary Fig. 7). These results indicate that, after the CE initiates, the kinetic balance between anisotropic CE progression, $\text{Cu}_{1.8}\text{S}$ etching and MnS deposition induces the specific transformation of hexagonal NPLs into crescent-shaped HNPLs, as summarized in Fig. 3n.

Q3: On page 10 the authors state that anisotropic CE progression and $\text{Cu}_{1.8}\text{S}$ etching by Cl^- occur simultaneously to induce the transformation of hexagonal $\text{Cu}_{1.8}\text{S}$ NPLs into crescent-shaped $\text{Cu}_{1.8}\text{S}$ -MnS HNPLs. However, as discussed above this is inconsistent with the data presented and can be better explained by a kinetic imbalance.

Author reply: As we addressed above, we added the MnS deposition process to the originally proposed mechanism.

Q4: On page 11 the authors invoke the nanoscale Kirkendall effect to explain the formation of the crescent-shaped $\text{Cu}_{1.8}\text{S}$ -MnS HNPLs (i.e., the “waning” phase). However, this seems unlikely as the Kirkendall effect should rather result in the formation of hollow NPLs (i.e., dissolution of the center of the NPLs), as previously reported by Buhro and coworkers for CuInS_2 NCs (ref. [18]). To support this interpretation, the authors should also propose a reasonable explanation for the asymmetry observed in the process: why does one half of the NPL dissolves, while a MnS shell overgrows on the other? It seems more likely to me that the dissolution is driven by a kinetic

imbalance in between etching rates by Cl^- and the CE rates (similarly to the formation of hollow Cu_{2-x}S nanocrystals through etching induced by GaCl_3 [Hinterding, et al.; ACS Nano (2019), 13, 12880]).

Author reply: As we addressed above, the combination of CE, Kirkendall-type deformation and etching of $\text{Cu}_{1.8}\text{S}$ plays an important role in the revised mechanism of anisotropic shape changes in this study. A decisive factor in determining the final shape depends on where the CE initiates. Initiation of CE from a single location on a NC forms the starting point for the subsequent CE, often leading to the formation of Janus-type heterostructure, as observed in many cases [refs. 8, 26–34]. Explanations for this phenomenon have been often provided by the formation of a crystallographically stable heterointerface and/or the presence of a high activation energy for the CE reaction. These explanations would also apply to our case, where CE of the $\text{Cu}_{1.8}\text{S}$ NPL with Mn^{2+} started from a single location, and subsequently an unstable surface is exposed on the opposite side as a result of deformation due to the imbalance in cation diffusion rates. This exposure accelerates etching at that side, resulting in the formation of the crescent-shaped plate. Although we do not have clear answer to the question why the CE reaction starts at a single position of NPL, as we showed in the previous paper (*Science* **2021**, 373, 332.), the position where the CE reaction starts is greatly dependent on the NC shape.

Q5: The authors convincingly demonstrate that chloride is responsible for the etching and dissolution observed during the waning phase. However, metal chlorides are widely used to deliver guest cations in cation exchange reactions and do not seem to induce any etching. Why do they act as etching agents under the conditions used in the present work? What is different with respect to the conditions typically used in other works?

Author reply: As you pointed out, metal chloride is widely used in many CE studies, and we can rarely find the severe etching and dissolution. In numerous cases, the NCs are inherently surrounded by thermodynamically stable crystal facets that are effectively protected by ligands, preventing Cl^- and/or TOP from attacking. However, if outward/inward cation diffusion is imbalanced, it causes a change in NCs' shape to destabilize stable facets. Consequently, the exposed unstable surfaces are preferentially attacked and etched by Cl^- and/or TOP. Since partial CE reactions of NPLs or nanorods with Mn^{2+} are quite rare, our results seem to be specific to the NC shape. Because we can find few reports on the details of size and volume changes NCs after partial CE reactions, the effect of etching observed in this study might have been overlooked.

Q6: The discussion on page 11 and elsewhere in the paper overlooks the importance of the solid-state diffusion rates of the guest cations in determining the overall heteroarchitecture of the product nanocrystals obtained by cation exchange reactions. This is a crucial point because it depends on the nature of both the guest cation and the host nanocrystal and on the temperature. Slower migration of the incoming guest cation does not necessarily result in collapse of the anionic sublattice and dissolution of the NC, but rather in the formation of heteronanocrystals with well-defined heterointerfaces, while fast diffusion results in homogeneous alloys. This has been extensively investigated by many groups, e.g., Schaak and coworkers [refs. 30-32], Swihart and coworkers [Chem. Mater. (2018), 30, 1399], Robinson and coworkers [Nano Lett. (2014), 14, 7090], Donega and coworkers [Groeneveld, E.; et al. ACS Nano (2013), 7, 7913; Hinterding, et al.; ACS Nano (2019), 13, 12880], Manna and coworkers [ref. 4].

Author reply: Thank you for your comments. In the references you showed, the relatively slow progression of CE reactions seems to be attributed to the low temperature or absence of TOP. For instance, in the research of Schaak et al. [refs. 32 and 42] and Robinson et al. [Nano Lett. 2014, 14, 7090.], partial CE was conducted at 50–90°C, potentially leading to weak etching. Additionally, detailed information about size and volume changes during the process is missing, which would raise the possibility of volume changes beyond lattice expansion/compression. Moreover, in the study by Swihart et al. [Chem. Mater. 2018, 30, 1399. (ref. 27)], attention is primarily focused on doping rather than CE, and the resolution of TEM images is not high. The NPLs seem to suffer damage or deformation. We think that a more careful observation dataset is necessary for deeper discussion. These works often lack insight into the difference in diffusion rates of cations. Difference in diffusion rates of guest and host cations is discussed in cases demonstrating the Kirkendall effect but seems to be overlooked in cases without observable shape changes. Even if the overall CE reaction rate were slow, deformation would not occur if the inward/outward diffusion rates of cations are comparable.

Comparing various precedents with vastly different combinations of ionic crystal species, precursors, temperatures, concentrations etc., is not straightforward, and systematizing such information is rather challenging. In this study, we combined discussions from the previous literature on CE leading to transformation with the data of actual temporal evolution of heterostructures from controlled experiments under comparable conditions. We believe that the revised mechanism proposed in Authors reply to Q2 is clear enough.

Q7: Page 12, line 230: “Fig. 4” should be “Fig. 5”.

Author reply: Thank you for your careful reading.

Manuscript revision:

We revised the Figure number in Page 13.

Q8: The mechanism proposed for the “waxing” process is rather vague and does not seem to be fully supported by the data presented. The cartoon in Figure 5 assumes that the starting configuration for the waxing process corresponds to the end configuration of the waning process. This is however not supported by the TEM images and STEM-EDX maps presented in Figure 5, which are consistent with the intermediate configuration shown in the cartoon in panel 1 of figure 5 but provide no evidence for the starting configuration. Moreover, the mechanism proposed leaves many intriguing questions without satisfactory answers, e.g.:

(a) why would the MnS shell formed during the waning phase suddenly redissolve and regrow to regenerate the missing half of the nanoplate? The conditions are clearly favorable to MnS growth, so why would they at the same time promote the dissolution of what was seemingly a very stable MnS shell?

(b) And why would the thickness of the NPL revert back to the original thickness? What prevents the CE to proceed and convert the $\text{Cu}_{1.8}\text{S}$ core into MnS while additional MnS grow on the sides of NPLs concomitantly reconstructs the missing half of the nanoplate, yielding a thicker hexagonal NPL?

(c) Why does the $\text{Cu}_{1.8}\text{S}$ that was stored in the core of the core-shell $\text{Cu}_{1.8}\text{S}$ -MnS heteroNPLs suddenly migrates to the surface and spreads itself over just one of the wide facets? The authors are either missing several crucial intermediate steps in the conversion of the left cartoon to the middle cartoon or are mistaken in their mechanism proposed for the waxing process.

Author reply: The experiments in Fig. 5 illustrate the temporal evolution of NPLs after the addition of extra MnCl_2 to the $\text{Cu}_{1.8}\text{S}$ -MnS HNPLs ($\text{MnCl}_2/\text{Cu}_{1.8}\text{S} = 0.5$). The starting configuration before the addition of MnCl_2 was checked by sampling a small amount of the reaction solution (Fig. R9), which corresponds to the samples at 0 s in Fig. 5. The starting samples show the crescent-shaped HNPLs, which are consistent with the products in Fig. 2, Fig. 3e formed by partial CE and the left model in Fig. 5l. Figures 5a and 5e–k represent the products formed at 30 s after the addition of extra Mn^{2+} , corresponding to the middle model in Figure 5l. These products (30 s) possess neither crescent- nor complete plate-shaped structure, suggesting the formation of intermediates during the waxing process.

Figure R9. TEM image of $\text{Cu}_{1.8}\text{S}$ - MnS HNPLs before addition of MnCl_2 , corresponding to the samples at 0 s in Fig. 5.

(a) We suggest that MnS shell is not re-dissolved but Mn^{2+} ions exchange position with Cu^+ ions through ion diffusion within the HNPLs. The schematic in Fig. 5l represents a continuous incorporation of Mn^{2+} (via both CE and deposition). MnS deposition continuously occurs from waning step, as discussed in Author reply to Q2.

(b) We suggest two scenarios of how the thickness of the NPL reverts back to the original thickness. 1) If CE proceeds in the $\text{Cu}_{1.8}\text{S}$ layer on MnS after 30 s, a thick MnS plate will form exclusively in that region (Fig. R10a). Considering the formation of flat, thin MnS NPLs in our experiments, the shape reconstruction should occur to make the thickness uniform after the CE. Completely flat NPLs seem more stable due to the reduced surface energy than those with uneven thickness.

2) The $\text{Cu}_{1.8}\text{S}$ layer is rapidly etched by Cl^- and/or TOP, leaving a thin MnS layer and releasing S^{2-} as the precursor for MnS growth (Fig. R10b).

For both cases, the string part of crescent-shaped HNPL is preferentially repaired by MnS deposition due to its unstable nature of the damaged structure (Fig. R11). Because the thicknesses of MnS parts in bilayer structure (10 s in Fig. 5) are slightly thinner (~ 4.7 nm) than that of final MnS NPLs (5.2 nm) (Fig. R12), it seems that both processes may occur rather than just one process or the other.

Figure R10. Schematic of conversion of $\text{Cu}_{1.8}\text{S}$ into MnS in the final stage of CE from the side view: (a) homogenizing of uneven thickness after CE of $\text{Cu}_{1.8}\text{S}$ layer; (b) repairing damaged part with MnS after etching of $\text{Cu}_{1.8}\text{S}$ layer.

Figure R11. Enlarged HAADF-STEM image of the crescent-shaped $\text{Cu}_{1.8}\text{S}$ - MnS HNPLs in Fig. 2a.

Figure R12. HAADF-STEM image of $\text{Cu}_{1.8}\text{S}$ - MnS HNPLs at 10 s in the waxing process with total and MnS layer thicknesses.

(c) Thank you for your comments. Because the transformation from the crescent-shaped NCs to the intermediate NCs with bilayer structure (at 30 sec) involves a large structural change, there should be other intermediate structures in the earlier stage. However, the waxing stage progresses quite rapidly, making it difficult to experimentally capture fine intermediate snapshots. In previous works, reconstruction of heterointerfaces in NCs has been often observed in partial CE reactions. For example, in the case of $\text{Cu}_{2-x}\text{S}/\text{CuInS}_2$ [C. Xia et al., *ACS Nano* **2021**, *15*, 9987. (ref.36)], a single flat heterointerface was formed from the initial heterointerface at different positions. The large mobility of cations in $\text{Cu}_{1.8}\text{S}$ especially under heated conditions can rearrange two distinct domains with multiple patchy structure [H. L. Young et al. *JACS* 2023, *145*, 23321. (ref. 40)] or core-shell structure [R. Tu et al. *JACS* 2016, *138*, 7082. (ref. 31)] into a smaller number (area) of heterointerfaces (Fig. R13). These works suggest that a more crystallographically stable $\text{Cu}_{1.8}\text{S}$ – MnS heterointerface is spontaneously formed during the CE progression in our case. Zhou et al. reported that Cu_{2-x}S – MnS heterostructured NCs tend to form only one heterointerface [J. Zhou et al. *CrystEngComm*, 2013,*15*, 4217. (ref. 41)]. The difficulty in growing on both planes of Cu_2S NPLs also suggests the thermodynamic stability of $\text{Cu}_{1.8}\text{S}$ – MnS bilayer structure.

[Redacted]

H. L. Young et al. *J. Am. Chem. Soc.* 2023, *145*, 23321.

Figure R13. Reconstruction of heterointerfaces in partially cation-exchanged NCs. (Reprinted with permission from *J. Am. Chem. Soc.* **2023**, *145*, 23321. Copyright 2023 American Chemical Society.)

Manuscript revision:

To clearly explain the transformation mechanism of waxing process to readers, the above discussion on the waxing process has been partly added to the main text. Figures R10 and R12 were added as Supplementary Fig. 14 and 15, respectively.

Page 14, 2nd paragraph:

Because the transformation from the crescent-shaped to the intermediate NCs with bilayer structure (at 30 sec) involves a large structural change, there should be other intermediate structures in the earlier stage. However, the waxing stage progresses quite rapidly, making it difficult to experimentally capture fine intermediate snapshots. In previous works, reconstruction of heterointerfaces in NCs has been often observed in partial CE reactions. The large mobility of cations in $\text{Cu}_{1.8}\text{S}$ especially under heated conditions can rearrange two distinct domains with multiple patchy structure⁴⁰ or core-shell structure³¹ into a smaller number (area) of heterointerface. These works suggest that, in our case, the $\text{Cu}_{1.8}\text{S}$ –MnS bilayer structure is spontaneously formed by generating more thermodynamically stable heterointerface from the $\text{Cu}_{1.8}\text{S}@MnS$ core@shell structure during CE progression⁴¹.

Page 15, 3rd paragraph:

A distinctive phenomenon in the waxing process is the decrease in the NPL thickness when the $\text{Cu}_{1.8}\text{S}$ –MnS bilayer structure (7.5 nm at 10 s) is transformed into MnS NPL (5.2 nm). We suggest two possible scenarios of how the thickness of the NPL reverts back to the original thickness. The first is based on the reconstruction process after the CE reaction. If the CE proceeds in the $\text{Cu}_{1.8}\text{S}$ layer of $\text{Cu}_{1.8}\text{S}$ –MnS bilayer structure, a thick MnS plate will form exclusively in that region, resulting in the formation of MnS NPLs with uneven thickness (Supplementary Fig. 14a). Considering the formation of flat, thin MnS NPLs, the shape reconstruction should take place to make the thickness uniform after the CE. Completely flat NPLs seem more stable due to the reduced surface energy than those with uneven thickness. In another scenario based on the etching and deposition process, the $\text{Cu}_{1.8}\text{S}$ layer of $\text{Cu}_{1.8}\text{S}$ –MnS bilayer structure is rapidly etched by Cl^- and/or TOP, leaving a thin MnS layer and releasing S^{2-} as the precursor for MnS growth (Supplementary Fig. 14b). Because the thickness of MnS parts in bilayer structure (10 s in Fig. 5) is slightly thinner (~4.7 nm) than that of final MnS NPLs (5.2 nm) (Supplementary Fig. 15), the two scenarios may occur simultaneously rather than just one process or the other.

Q9: In the final section of the paper the authors discuss their experiments with nanorods. How do their observations differ from what has already been extensively reported by Schaak and coworkers? (e.g., refs. [30-32])

Author reply: Our experiments with nanorods were conducted to confirm the versatility of the waning-and-waxing process in the CE reactions. In other studies on partial CE of nanorods using ZnCl_2 precursor (refs. 32 and 42) the CE reactions were performed at 50°C and 90°C, respectively, which are lower than in our study, suggesting that the etching process may not be activated (since higher reaction temperatures accelerate the etching of $\text{Cu}_{1.8}\text{S}$ NCs [G. A. Di Domizio et al., *Chem.*

Mater. 2021, 33, 3936. (ref. 37)]). Another noticeable aspect in comparison with other studies is the detailed investigation of the changes in width, length, and volume of partially cation-exchanged nanorods (as well as NPLs). Structural changes beyond lattice volume changes were found through such detailed investigation, and also the novel phenomena such as partial etching were observed, which have been overlooked.

Q10: Claim of universality at the end of paper is likely exaggerated. The authors investigated four different cations: Mn^{2+} , Zn^{2+} , Cd^{2+} and Fe^{3+} . The waning-waxing cycle process occurs only for Mn^{2+} and Zn^{2+} . Cd^{2+} gives Janus-type hetero-NPLs and Fe^{3+} gives homogeneous CuFeS_2 . What determines the different outcomes? Likely the different solid-state diffusion rates of the different cations, but this is an aspect that is largely overlooked in the mechanism proposed in the paper.

Author reply: As you pointed out, the waning process was observed only for Mn^{2+} and Zn^{2+} , limiting the range of applicable metal cations in our experimental conditions. In the experiments in Fig. 6, partial CE of $\text{Cu}_{1.8}\text{S}$ NPLs was conducted with Mn^{2+} or Zn^{2+} at first, and the subsequent CE reactions of chipped $\text{Cu}_{1.8}\text{S}$ –(MnS or ZnS) HNPLs were performed with four different cations (waxing process). We expected that the waxing process might occur even with different metal cations based on the assumption that the waxing process involves both the CE of the remaining $\text{Cu}_{1.8}\text{S}$ and the deposition of metal sulfide formed from supplied metal cations and dissolved S^{2-} , as observed in the case of Mn. As a result, the overall hexagonal-plate shapes are restored with any metal cations (Fig. 6), suggesting the potential use of various metal cations in the waxing process. We think that the different outcomes are probably derived from the thermodynamic stability of the resulting crystal phases (CuFeS_2 phase is formed easier than Fe_2S_3 phase). Therefore, we believe that the waxing process is general and versatile.

Manuscript revision:

To precisely conclude the manuscript, we revised the summary as follows.

Page 19, 2nd paragraph:

We have also shown that the waning and waxing strategy can be induced by other metal cations and can be applied to NRs and that the waxing cycle is a general and versatile process.

Reviewer #1 (Remarks to the Author):

I believe that the authors have adequately addressed the reviewers' questions and concerns, and thus recommend acceptance.

Reviewer #2 (Remarks to the Author):

The authors have thoroughly addressed my comments. Now, I recommend that this manuscript be published as it is.

Reviewer #3 (Remarks to the Author):

Reviewer #4 (Remarks to the Author):

The authors have adequately addressed all my concerns. Therefore, I think the revised version of the manuscript merits publication in Nature Communications.

Manuscript ID: NCOMMS-23-46418-T

Title: Waning-and-waxing shape change in ionic nanoplates upon cation exchange

Authors: Zhanzhao Li, Masaki Saruyama, Toru Asaka and Toshiharu Teranishi

Author replies to the comments of Reviewer 1

I believe that the authors have adequately addressed the reviewers' questions and concerns, and thus recommend acceptance.

Author reply: We are grateful for your positive recommendation.

Author replies to the comments of Reviewer 2

The authors have thoroughly addressed my comments. Now, I recommend that this manuscript be published as it is.

Author reply: We are grateful for your positive recommendation.

Author replies to the comments of Reviewer 3

Author reply: We are grateful for your positive recommendation.

Author replies to the comments of Reviewer 4

The authors have adequately addressed all my concerns. Therefore, I think the revised version of the manuscript merits publication in Nature Communications.

Author reply: We are grateful for your positive recommendation.